

# Machine learning-driven characterization and prescription of aerosol optical properties for atmospheric models

Nilton Évora do Rosário[1], Karla M. Longo[2], Pedro H. Toso[1], Saulo R. Freitas[2], Marcia A. Yamasoe[3], Luiz Flávio Rodrigues[2], Otavio Medeiros[2], Haroldo Campos Velho[2], Isilda da Cunha Menezes[4], Ana Isabel Miranda[4]

[1] Departamento de Ciências Ambientais, Universidade Federal de São Paulo, Diadema, SP Brazil

[2]Instituto Nacional de Pesquisas Espaciais (INPE), São José dos Campos, SP, Brazil

[3]Departamento de Ciências Atmosféricas, Instituto de Astronomia, Geofísica e Ciências Atmosféricas, Universidade de São Paulo, Cidade Universitária, São Paulo, SP, Brazil

[4]Center for Environmental and Marine Studies (CESAM), Department of Environment and Planning, University of Aveiro, Campus Universitário de Santiago, 3810-193 Aveiro, Portugal

Correspondence to: Nilton do Rosário (nrosario@unifesp.br)

## Abstract

Accurate modeling of aerosol optical properties is critical to simulate aerosol radiative effects. However, uncertainties regarding the simulation aerosol intensive optical properties are still significant. Therefore, the use of observations to constrain aerosol optical properties in models has been indicated as an option. Also, explicit computations of optical properties are still too costly for operational models, which make observational-based prescriptions a convenient solution. We developed a observational-based prescription of aerosol optical properties driven by machine-learning techniques that can be applied in models. The Iberian Peninsula (IP) was taken as the reference domain, and the aerosol products from the AERONET sites across the IP as the main dataset. First, clustering was applied to define the typical aerosol optical regimes affecting the IP atmosphere. Five typical regimes were identified. Two of them were dominated by coarse mode, which were associated with Saharan dust. One was found to be close to pure dust, while the other indicated a mixed scenario of dust and pollution. Two of the non-dust regimes, strongly and moderately absorbing, were found to be associated with smoke. The remaining non-dust regime, with not a clear association, occurs mostly in the eastern portion of the IP. Afterward, using aerosol-type columnar mass density from MERRA-2, a model was trained as predictor of the optical regimes using the Random Forest method. The model was tested under distinct aerosol scenarios. Predictions' accuracy ranged from 60 to 75%, depending on the regime, while presenting an average accuracy of 70%.

**Keywords**: Aerosol Optical Properties, AERONET, MERRA-2, Machine-Learning, Random Forest





## 1. Introduction

Aerosol particles' importance in the Earth's climate system is undisputed. Via the scattering and absorption of terrestrial and solar radiation, aerosol particles are direct players in the planetary energy budgets (Kim and Ramanathan 2008, IPCC, 2021, Li et al., 2022). However, this role is permeated by high complexity and significant uncertainty (Spencer et al. 2019, IPCC, 2021, Li et al., 2022). The uncertainties and challenges in accurately representing aerosol particles' processes in climate, weather, and environmental models arise from various limitations. For instance, focusing on aspects related to the direct interaction with radiation, limitations in the current global observational system to address information such as spectral complex refractive index and size distribution, two critical microphysical variables to the characterization of particle absorption and scattering (Samset et al. 2018, Li et al., 2022), are still a relevant source of uncertainty.

The lack of geographical representativity of the traditional libraries of aerosol optical and microphysical properties (Shettle and Fenn, 1979, Koepke et al., 1997, Hess et al., 1998) has been central in the aerosol optical properties uncertainty debate. Another critical aspect is the characterization of the state of the mixture of the aerosol particles in the model's aerosol modules (Samset et al. 2018, Sand et al., 2021). Given the complex dynamic of aerosol particle emission, transport, and removal in the atmosphere, numerical modelling of the state of the mixture and the resultant complex refractive index and size distribution is widely recognized as one of the most important sources of uncertainty in addressing aerosol particles' radiative forcing (Sand et al., 2021). According to Sand et al. (2021) aerosol absorption is poorly constrained, and the current climate models present a large range in the quantification of the main absorbing aerosol species (black carbon (BC), organic aerosols (OA), and mineral dust). Brown et al. (2021) findings indicate that biomass-burning aerosols in most climate models are too absorbing mainly due to treatments of aerosol mixing state. Saharan dust, a critical component of the global aerosol system, has been found to absorb less solar radiation than models estimate (Adebiyi et al., 2023), and the primary cause pointed out is the models overestimate of the dust imaginary refractive index. Absorption is not the only issue facing aerosol particle representation in climate models, the relative contribution of fine and coarse mode particles is also a challenge. For instance, Adebiyi et al. (2023) also found models underestimating large dust particles when representing North African dust plumes.

Observation-constrained models have been recommended to mitigate models' current difficulty in fully simulating aerosol properties and processes accurately (Samset et al. 2018, Proske et al., 2024). In addition to the uncertainty aspects, explicit simulation of aerosol compositions and microphysical, followed by explicit computations of intensive optical properties, is still too expensive computationally for operational models, which also makes observational-based prescriptions a convenient solution. Zhong et al. (2022) used relationships from an ensemble of aerosol models and satellite observations to identify the primary source of uncertainty in aerosol modelling results. Their study pointed out the incorrect lifetimes and the underestimation of mass extinction coefficients as the most critical drivers of bias in aerosol simulations. As the largest, time and device consistent observational network, capable of constraining multiple aerosol intensive microphysical and optical properties, the AErosol RObotic NETwork (AERONET) has been used worldwide to



constrain models and satellite algorithms (Omar et al., 2005, Li et al., 2010, Levy et al., 2010,
Rosario et al., 2013, Russel et al., 2014, Chen et al., 2023). Chen et al. (2023) developed an
aerosol optical module with observation-constrained Black Carbon properties to improve
aerosol absorption simulation. Their sensitivity simulations show a reduction of 18%–69%
in the biases of aerosol single-scattering co-albedo when compared with global observations
from AERONET. Li et al. (2010) used AERONET retrievals to evaluate and improve the
performance of a GCM aerosol optical module. They found their GCM to simulate flatter
Aerosol Optical Depth (AOD) spectral dependence, indicating an Angstrom Exponent (AE)
biased to low values, which suggests that the aerosol sizes simulated were too large. After
adjusting the aerosol's size based on AERONET retrievals the agreement between simulated
and observed AOD improved for all aerosol regimes, but especially for smoke and dust
scenarios. Rosario et al. (2013) used a set of spectral optical models developed from
AERONET sky retrievals over distinct biomes combined with the concept of anisotropic areas
of influence of the AERONET sites (Hoelzemann et al., 2009) to constrain smoke aerosol
radiative effect modelling during South America biomass burning. By doing so, they were
able to capture the effect of the regional variability of smoke optical properties (absorption
and size related) on the surface solar irradiance related to the biomes' distinct nature of
smoke.
Global and regional cluster analysis of AERONET long-term retrievals of aerosol properties
has proved valuable to classify observations in terms of aerosol optical regimes, providing
means to qualitative constraints on aerosol properties (Omar et al, 2005, Levy et al., 2007,
Russell et al., 2014, Li et al., 2019, Fan et al., 2020, Zhou et al., 2023). In these studies, the
number of the identified typical aerosol optical regimes varied from 4 to 10, numbers that
were expected to likely represent either global or regional major aerosol scenarios
variability, according to each study focus. In their study, Zhou et al. (2023) found regional
aerosol regime classifications to perform better than global classifications when applied to
simulate AOD during pollution episodes and in different seasons in Beijing, China. They found
a large difference between the strongly and moderately absorbing aerosol regimes in the
global and regional clustering results. Two major sources of differences between their global
and regional clustering for China were aerosol optical regimes dominated by dust and smoke
particles. Compared to China, Zhou et al. (2023) pointed out that smoke and dust-dominated
optical regimes are more common globally. Their result suggests that regional classification
better captures typical aerosol optical regimes influencing a specific domain and, therefore,
with potential to improve observation-constrained simulations of aerosol radiative forcing.
Focusing on the Iberian Peninsula (IP), this study sought to characterize the typical aerosol
optical regimes driving the variability of aerosol-intensive properties over the peninsula,
aiming to constrain aerosol optical properties prescription in atmospheric models using a
novel approach based on machine-learning approach. IP is a region affected by a highly
dynamic and complex set of aerosol mixing, including natural and anthropogenic particles
(Cachorro et al., 2016, Gomez-Amo et al., 2017). Natural sources include marine aerosols
from the Atlantic Ocean and Mediterranean Sea, mineral dust from North Africa, and
eventually, wildfire emissions. Major anthropogenic sources are urban-industrial,
particularly in more densely populated regions, and biomass burning driven by human
activities, especially in the north and central Portugal and eastern and north of Spain.



Regional column-integrated optical properties are highly sensitive to the mixing of this
diversity of aerosol-types, in particular to dust and smoke mixing (Gomez-Amo et al., 2017).
The manuscript is organized as follows: Section 2 includes a brief overview of the Iberian
Peninsula, focusing on the main atmospheric circulation features and major aerosol particle
sources affecting the region, followed by the description of the dataset and methods adopted
to identify, characterize and prescribe the identified aerosol typical regimes. Results and
discussions are presented in Section 3. First, the identified aerosol optical regimes and their
major features are described and contextualized. Subsequently, the results of the novel
machine-learning approach to prescribing the optical regimes are discussed and evaluated.
Finally, the main findings of our study are highlighted in the conclusion section.

## 134 **2. Study Region, Data and Methods**

### 136 **2.1 Study region**

The Iberian Peninsula (**Figure 1**), comprising Spain and Portugal, exhibits diverse climate
conditions due to its complex topography and proximity to the Atlantic Ocean, the
Mediterranean Sea and North Africa. The wind circulation over the peninsula is shaped by its
location between the Atlantic Ocean and the Mediterranean Sea, diverse topography, and
interactions between regional and global atmospheric patterns, leading to complex wind
circulations that significantly influence the region's climate. This results in distinct climate
zones, from arid deserts to lush green forests. The Mediterranean climate spans most of
Spain, including the eastern and southern coastal regions and central Portugal, featuring hot
and dry summers, especially inland.  Winters are mild, rarely dropping below 10°C in coastal
areas. Most precipitation, often rain, occurs in autumn and winter, leading to dry summers
that increase wildfire risks. Wildfires regularly occur in the IP region fueled by extreme
weather conditions, abnormal high temperature records combined with strong, dry winds
(Asfaw et al., 2022, Ermitão et al., 2023). Under these scenarios the entire region can be
affected by smoke plumes that often shape the entire region's optical properties (Elias et al,
2004, Gomez-Amo et al., 2017). But wildfires are more frequent in the north and central
region of Portugal and the north and eastern portion of Spain (Ermitão et al., 2023, Alvares
et al., 2024). Oceanic climate is typical in northern coastal regions of Spain, such as Galicia,
Asturias, and the Basque Country, and parts of northern Portugal. The Atlantic Ocean
influences mild temperatures year-round, with minimal seasonal variation and abundant,
evenly distributed rainfall. Annual precipitation can exceed 1,000 mm, with frequent cloud
cover and high humidity, especially in winter.  The Continental climate of the central plateau
(Meseta Central) and the Ebro Valley features extreme temperature variations, with hot
summer, highs often above 35°C, and winter below freezing. The central regions have less
precipitation than the coastal areas, with a semi-arid climate in some parts. Most rainfall
occurs in spring and autumn. Arid and Semi-Arid Climates are found in Southeastern Spain,
especially in Murcia and Almería, and parts of the Ebro Valley. These areas receive very low
rainfall, often less than 300 mm annually, leading to desert-like conditions like those in the
Tabernas Desert. Summers are extremely hot, while winters are mild. Southern Spain,



especially the Andalusian region, can be affected by hot and dry winds from the Sahara,
causing heat waves and dust storms.
The occurrence of Saharan dust events on Iberian Peninsula usually peaks in March and June,
with a marked minimum in April and lowest occurrence in winter according to Cachorro et
al. (2016). Depending on the synoptic conditions and circulation patterns, dust transport can
affect the entire peninsula (Toledano et al., 2007). The prevailing westerlies, blowing from
west to east, are the dominant wind pattern over the Iberian Peninsula. These winds are most
prominent in the mid-latitudes, including the Iberian Peninsula. More pronounced in the
northern region during autumn and winter, these winds bring moist air from the Atlantic,
increasing precipitation in Galicia, the Basque Country, and north Portugal. While they also
affect central and southern areas, their impact is moderated by the peninsula's topography
and other wind systems. The northeast trade winds affect the southern and western coasts
of Portugal and southwestern Spain, creating a mild and dry climate, especially in summer.
In contrast, Mediterranean winds affect the eastern and southeastern coasts. Additionally,
the Iberian Thermal Low, resulting from intense heating of the Iberian interior, creates a low-
pressure area that draws air from the Atlantic and Mediterranean shores, leading to
converging wind patterns. This circulation pattern enhances sea breeze penetration and
moderates coastal temperatures. Southern Spain is influenced by Sahara winds, as said these
dry winds often carry dust, increasing temperature and reducing air quality. Calima is a type
of wind that occurs when Saharan dust reaches the peninsula, especially in summer, causing
hazy skies, a reddish tint, and low visibility. These winds are linked to high-pressure systems
over North Africa and low-pressure systems over the western Mediterranean.
The wind circulation over the Iberian Peninsula is a dynamic and complex system shaped by
global atmospheric patterns, regional geography, and local topography. The interaction of
prevailing westerlies, trade winds, Mediterranean breezes, and local wind systems creates a
diverse wind regime that affects the peninsula's climate. Understanding these patterns is
essential for weather prediction, agriculture management, and tackling environmental
challenges. According to Cachorro et al. (2016), these complex and contrasting influences of
air masses from the Atlantic Ocean, Mediterranean Sea, European continent, and North Africa
lead to a large spatio-temporal variability in aerosol properties, types, and mixing processes
over the Iberian Peninsula. This makes the peninsula a challenging region for online
modeling of aerosol microphysical properties and mixing state, therefore an interesting
region to evaluate observation-constrained approaches.



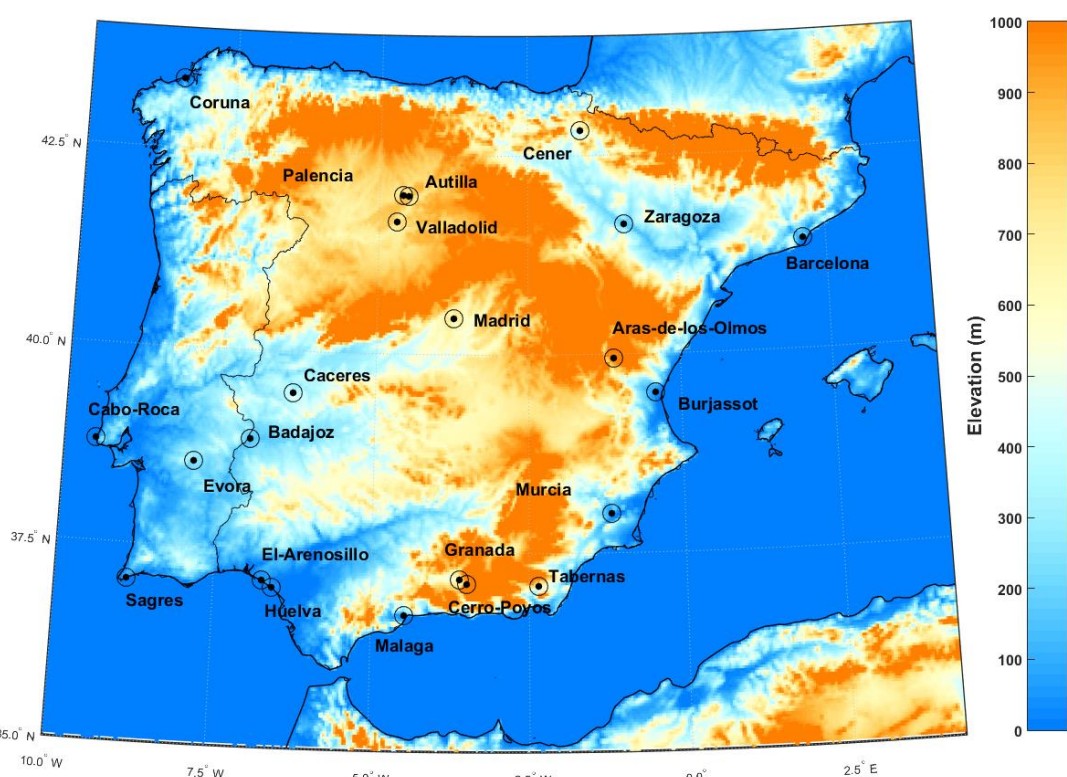

**Fig. 1**: *AERONET sites locations displayed on top of the Iberian Peninsula topography.*

## 2.2 AERONET aerosol inversion product

AERONET is a global ground-based network of sun photometers mainly aimed at characterizing columnar aerosol particles properties (Holben et al., 1998). From the direct Sun attenuation measurements, AERONET algorithms derive spectral Aerosol Optical Depth ($AOD_\lambda$) at the wavelengths 0.34, 0.38, 0.44, 0.50, 0.67, 0.87, 0.94, and 1.02 µm. From the spectral dependency of AOD at these wavelengths, AERONET provides Angstrom Exponent (AE), a parameter sensitive to the aerosol particle size distribution (Eck et al., 1999). But for the present study, AERONET also provides several other intensive properties that depend not on the amount but on the nature of the aerosol, related to particle size, shape, and composition, from sky radiance measurements at the wavelengths 0.44, 0.67, 0.87, and 1.02 µm(Sinyuk et al., 2020). These intensive properties include microphysical parameters, such as refractive indices ($n+ik$) and volume size distribution, and also optical parameters like Single Scattering Albedo (SSA), asymmetry parameter (ASY), Lidar Ratio (LR), Linear Depolarization Ratio (LDR), Angstrom Exponent among others (Holben et al., 1998, Dubovik et al. 2002). Given the dependency of these intensive properties on the aerosol type and mixture state, it is possible to characterize the aerosol scenarios over a specific AERONET site in terms of their nature and sources (Eck et al., 1999; Dubovik et al., 2002). Therefore,





with a well-distributed regional network of AERONET' sun photometers, as that covering
Iberian Peninsula, one can characterize the spatial dynamic of aerosol types and mixture
state influencing the regional aerosol regimes.
Three key aspects of aerosol nature have been widely used to link aerosol regimes with
particle emission sources. These aspects are absorption efficiency, size distribution and
shape (Dubovik et al., 2002). For instance, combustion-based sources, including biomass and
fossil fuel burning, produce aerosol dominated by fine mode particles, and absorption ranges
from moderate to strong, depending on biomass burning nature, fossil fuel and ageing
processes. In contrast, natural sources, such as deserts and marine environments, produce
aerosols dominated by coarse-mode particles. Marine aerosol particles are characterized by
very low absorption, while dust aerosol can exhibit high absorption, mainly in the UV and VIS
bands (Smirnov et al., 2002; Dubovik et al. 2002). Furthermore, the irregular shape of dust
particles is a key factor that differentiates them from other aerosol types. This distinctive
feature is captured by AERONET retrievals of the LDR (Shin et al., 2018). Source attribution
provides valuable insights into the typical intensive optical properties affecting the
atmospheric column of a site resulting from complex aerosol state mixtures. This
understanding is crucial as it addresses a major challenge that current aerosol modules in
climate models face. Reproducing climatological aerosol-intensive properties scenarios over
specific regions has been a major goal of atmospheric models. In addition to evaluating
aerosol modules in atmospheric models, AERONET´s optical properties typical regimes,
which can be expressed as spectral aerosol optical models (Omar et al., 2005; Levy et al.,
2007; Rosario et al., 2013; Zhou et al, 2023), are valuable for simulating aerosol direct
radiative effects in environmental models (Rosario et al., 2013, Li et al., 2019). This approach
is especially beneficial when/where high computational capacity is unavailable and explicit
aerosol modules are not feasible.
Aiming to identify a representative set of typical aerosol regimes that affect the Iberian
Peninsula, we applied cluster analysis methods (described in Sec. 2.4) to the AERONET
dataset, taking advantage of the extensive coverage of AERONET sites across the region.
**Table 1** presents a set of intensive properties provided by AERONET that was used to
identify typical aerosol scenarios in the Iberian Peninsula atmospheric column. The variables
displayed cover all the three previously mentioned aspects, absorption efficiency, size
distribution and shape, which are expected to characterize the distinct nature of aerosol
types and mixture anticipated in the study region. We selected only AERONET sites that
operated for at least two years and that have sky radiance inversion available with the
highest quality level 2.0. Some selected sites are still operational, while others have been
discontinued. **Figure 1** illustrates the geographical distribution of the chosen sites. Our
selection encompasses various landscapes of the Iberian Peninsula, from coastal plains
regions (Coruña, Sagres, Burjassot) to highland plateau in the interior (Madrid, Valladolid,
Aras-de-los-Olmos) and lowland valleys (Zaragoza, Murcia). Regarding external air mass
influence, sites in the southern border of IP are typically the first to experience the transport
of dusty air mass from North Africa, with locations such as El- Arenosillo, Huelva, Malaga,
Sagres affected. The eastern sites (Barcelona, Burjassot, Murcia) are expected to be strongly
influenced by the Mediterranean air masses. Western and northern sites (Cabo da Roca,
Coruna, Sagres) are directly under the influence of air mass from the Atlantic Ocean.





Additionally, Portugal countryside (Evora) and Spain eastern sites (Badajoz, Caceres) are
located in regions that very often experience biomass burning during the dry season
(Ermitão et al., 2023, Silva et al., 2023, Hammed e tal., 2024, Alvares et al., 2024).

**Table 1:** List of AERONET sky inversions intensive properties variables used in clustering
process.

| Variables | Abbreviation |
|---|---|
| Refractive Index - Real Part | $RI_{Real}(440)$, $RI_{Real}(670)$, $RI_{Real}(870)$, $RI_{Real}(1020)$ |
| Refractive Index - Imaginary part | $RI_{Imag}(440)$, $RI_{Imag}(670)$, $RI_{Imag}(870)$, $RI_{Imag}(1020)$ |
| Single Scattering Albedo | SSA(440), SSA(670), SSA(870), SSA(1020) |
| Asymmetry Parameter | ASY(440), SSA(670), SSA(870), SSA(1020) |
| Linear Depolarization ratio | LDR(440), LDR(670), LDR(870), LDR(1020) |
| Lidar Ratio | LR(440), LR(670), LR(870), LR(1020) |
| Fine and Coarse modes Volume median radius | VMR-F,VMR-C |
| Standard deviation from volume median radius, for Fine and Coarse modes | STD-F, STD-C |
| Fine and Coarse modes Effective radius | Reff-F, Reff-C |


## 2.3 Merra-2 Aerosol Diagnostic Product

The MERRA-2 (Modern-Era Retrospective Analysis for Research and Applications, Version 2)
Aerosol Diagnostic Product (ADP) is a comprehensive dataset provided by NASA that offers
global information about atmospheric aerosols (Gelaro et al., 2017, Buchart_Marchand et al.,
2017). MERRA-2 combines observational data with numerical models(reanalysis project) to
create a detailed long-term record of atmospheric dynamics and composition from 1980 to
the present. Among other variables, the MERRA-2 ADP product offers a long-term view of
aerosol mass distribution by types and the related optical properties (Buchart_Marchand et
al., 2017). Its extended temporal coverage allows analysis of aerosol trends, such as those
related to changes in atmospheric composition due to human activity and the impact on
climate. Key features of the MERRA-2 ADP include aerosol microphysical and optical
properties such as optical depth, mass concentration, and size distribution. These properties
are crucial for understanding aerosol loading and composition in the atmosphere and their
role in the Earth's radiation budget and climate system. A key aspect of MERRA-2 APD for
this study is that it provides aerosol-type column mass density, our target variable as a
predictor of aerosol optical model regime. The MERRA-2 APD includes diagnostics for the
aerosol types considered in most chemistry transport models: Dust (DT), Black-Carbon (BC),





Organic Carbon (OC), Sea-Salt (SS) and Sulfate (SF). The aerosol-type diagnostics variables
cover mass concentration at specific levels and integrated in the entire atmospheric column,
as well as columnar optical properties, such as extinction, scattering and absorption optical
depth. From these extensive aerosol-driven optical properties, it is possible to derive several
MERRA-2 ADP intensive optical properties, such as Single Scattering Albedo (SSA).
Given that the aerosol optical properties retrieved from each AERONET site are influenced
by the mixture of different aerosol types present in the local atmospheric column, it is
reasonable to assume that the impact of each aerosol type on the column's intensive optical
properties is primarily determined by its concentration. Based on this premise, we propose
a machine-learning approach that utilizes the aerosol-type column mass density predicted
by chemistry transport models to help us define the spatial distribution of the optical model
developed through cluster analysis of AERONET data. A description of the method presented
in this study, exploring MERRA-2 products, can be found in subsection 2.5.

## 2.4 Optical models development: Cluster Analysis

Cluster analysis has been extensively used to develop aerosol optical models based on
AERONET sky inversion products (Omar et al., 2005, Levy e al., 2007, Russel et al., 2014). The
underlying principle is that AERONET instantaneous retrievals can be grouped into a certain
number of clusters, each representing different categories of aerosol regimes. These studies
have explored mainly the K-means clustering method, one of the most popular unsupervised
machine learning algorithms for partitioning a dataset into a pre-defined number of clusters.
However, specifying the number of clusters in advance poses a significant challenge for the
K-means method. Fortunately, there are techniques available that minimize the subjectivity
involved in this pre-definition. In our study, we adopted the Elbow method (Shi et al., 2021),
the most widely used method for determining the optimal number of clusters (k) in a K-
Means clustering algorithm. It examines the relationship between the number of clusters and
the within-cluster sum of squares (WCSS), which measures the variance within each cluster
(**Eq. 1**)
$$WCSS = minimize(\textstyle\sum_{k=1}^{k} W(C_k)) \quad \textbf{(1)}$$
where $C_k$ is the $k$th cluster and $W(C_k)$ is the within-cluster variation. The total within-cluster
sum of squares (WCSS) measures the compactness of the clustering, and one wants it to be
as small as possible. We ran our clustering algorithm with k varying from 2 to 10 clusters.
For each $k$, we calculated the total within-cluster sum of squares (WCSS). The k results
against WCSS were displayed in a plot and the optimal number of clusters were defined based
on the location ($k$) of the bend (elbow) in the plot.

## 2.5 Optical models spacial prescription: Random Forest Technique

We propose a machine-learning approach that utilizes the well-known random forests
supervised algorithm (Breiman, 2001) to spatially represent the aerosol optical models
defined by the cluster analysis for each AERONET site (described in section 2.4). The



implemented method was tested using aerosol column mass density data from MERRA-2
(**Table 2**) to establish the spatial distribution of the optical regime defined by the clusters
average. This approach is also suitable for chemistry transport models.
MERRA-2 time series of column mass density for each aerosol type (DT, BC, OC, SS, SF) over
each AERONET site were collocated with the network inversion products used to derive the
clusters representing the distinct aerosol regimes over the Iberian Peninsula (described in
section 2.4). Each AERONET instantaneous aerosol microphysical and optical properties
inversion retrieval (Sinyuk et al., 2020) was connected to the corresponding cluster to which
it belonged. Likewise, each instantaneous aerosol microphysical and optical properties
inversion retrieval was also connected to the closest in-time combination of MERRA-2 data
of aerosol-type column mass density (DT, BC, OC, SS, SF). With this, we built a set of data fitted
to a training process, wherein the occurrence of each cluster (optical model) is related to a
particular mixture (combination) of aerosol types from MERRA-2 over each selected
AERONET site. The nature of our problem is classification, with the combination of the
aerosol-type columnar mass density, trying to predict which cluster of intensive optical
properties is more suitable for that particular combination.
Therefore, the first step was to split the data into training (70%) and test (30%). The
algorithm uses training data to learn the relationship between the combination of aerosol-
types columnar mass density and the target, which are the developed clusters from
AERONET aerosol-intensive properties. The training was done using the Random Forest
Classification algorithm (RandomForestClassifier) from the Python package Scikit-Learn
(Abraham et al. 2014). The Random Forest classifier's hyperparameters were optimized
using RandomizedSearchCV, a stochastic method of parameter space exploration. The
parameter space included the number of decision trees (n_estimators: 50–500) and the
maximum depth of trees (max_depth: 1–20). The process used stratified k-fold cross-
validation to ensure representative sampling across aerosol regime classes. This
optimization method addressed the issues of class imbalance and aerosol regime
classification in atmospheric measurements. The random search methodology was used to
find parameter combinations inside the parameter space without the processing demands of
grid search. Cross-validated performance indicators were used to select the final
configuration in order to reduce overfitting and ensure consistent performance across
aerosol regimes. The confusion Matrix was used to visualize the performance of the models,
and we also calculate the following indicators: Accuracy, Precision and Recall and F1
score.  Accuracy represents the number of correctly classified data instances over the total,
it checks the predictions against the actual values in the test set and returns the percentage
of times the model got right.
Precision and recall are two critical metrics for evaluating the performance of a classification
model. Precision is the proportion of true positives among all the predicted positive cases
(true and false), meaning it measures the accuracy of positive predictions (**Eq. 2**). Recall is
the proportion of true positives among all actual positive cases (true and false), meaning it
measures the model's ability to identify positive cases (**Eq. 3**). The F1 score, the harmonic
mean of a model's precision and recall, takes both precision and recall into account and
provides a more balanced measure of a model's performance (**Eq. 4**). The F1 score is set to





be a value between 0 and 1, indicating, respectively, poor precision and recall and high
precision and recall, which is ideal.

372                    Precision = True positive/(True positive + False positive) - **(2)**

373                    Recall = True positive/(True positive + False negative) - **(3)**

374                    F1 = 2 × (Precision × Recall)/(Recall + Precision) - **(4)**


**Table 2**: Predictor variables from Merra-2 (aerosol-type column mass density) used in the
machine learning process to prescribe the aerosol optical regime (optical model).

| Variables | Abbreviation | Unity | Spatial resolution |
|---|---|---|---|
| Dust column mass density | DUCMASS | kg/m^2 | $0.5° \times 0.625°$ |
| Black carbon column mass density | BCCMASS | kg/m^2 | $0.5° \times 0.625°$ |
| Organic carbon column mass density | OCCMASS | kg/m^2 | $0.5° \times 0.625°$ |
| $SO_2$ column mass density | SO2CMASS | kg/m^2 | $0.5° \times 0.625°$ |
| $SO_4$ column mass density | SO4CMASS | kg/m^2 | $0.5° \times 0.625°$ |
| Sea salt column mass density | SSCMASS | kg/m^2 | $0.5° \times 0.625°$ |

## 3. Results

The results section is divided into three subsections. The first one presents the results of
identifying the typical aerosol optical regimes affecting the Iberian Peninsula using cluster
analysis. The second subsection discusses the results and the performance of spatial
prescription of these typical aerosol regimes by applying machine learning (Random Forest)
to the columnar density of MERRA-2 aerosol components. Finally, case studies  applying the
method developed are presented and discussed.

### 3.1 Cluster Analysis: Optical models development

The number of clusters (*k*) selected to characterize the typical optical aerosol regimes over
the Iberian Peninsula was defined based on the Elbow method (**Figure 2**), which indicated
five as the optimal clusters number to capture the aerosol regime variability. We also
evaluated from the Elbow method that there is a sharp bending at *k=2*, which we associated
with a clustering separation between aerosol regimes strongly dominated by coarse mode,
dust regimes, and regimes dominated by fine mode, non-dust regimes. However, to cover
more specific regimes within these two macro-regimes (dust regimes vs non-dust regimes)
a higher *k* is required, and k=5 reveals to be the second sharpest bending. Cluster stability as





a function of the number of clusters was also evaluated as a complementary analysis. The
stability for k=5 is above the 90% threshold, similar to k=6, a number after which stability
sharply decreases. Therefore, combining the Elbow method and stability reinforced k=5 as
an optimal cluster number to capture the typical aerosol scenarios over the Iberian
Peninsula, reducing the subjectivity usually associated with the K-mean clustering method.

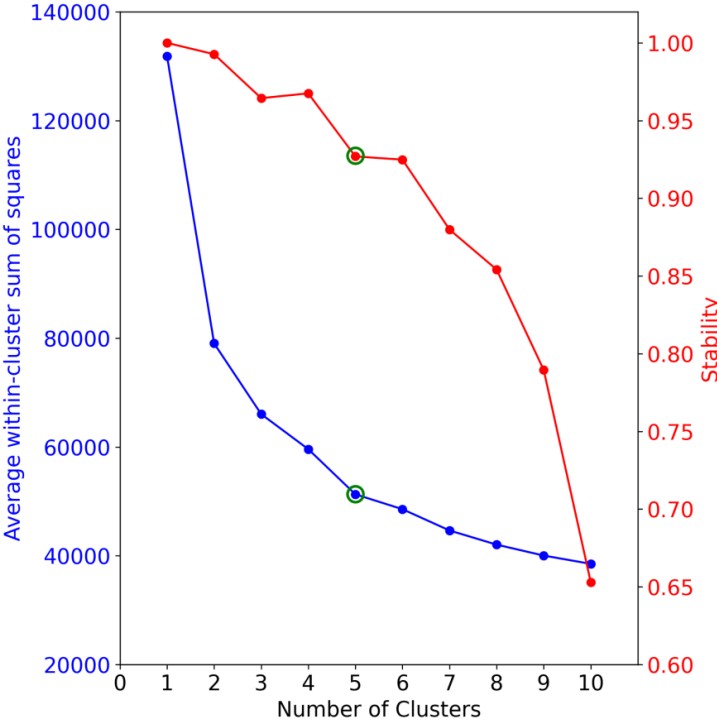


***Figure 2:*** *Average of sum of squares within-cluster and cluster stability as function of the*
*number of clusters.*

We applied the cluster analysis once we defined the optimal number of clusters. **Figure 3**
presents a combination of graphics used for aerosol properties analysis, highlighting the
obtained clusters' behavior and distinction. The first graphic (Fig. 3a) represents the Aerosol
Optical Depth (AOD) as a function of Angstrom Exponent (AE), which allows us to relate
aerosol loading variability with aerosol-regimes dominated either by coarse or fine mode
(Eck et al., 1999). This analysis shows that two of the clusters (C0 and C1) are regimes
dominated by coarse mode particles (AE < 1.0), while the remaining three (C2, C3, and C4)
are regimes under stronger influence of fine mode particles (AE > 1.0). The second plot
displays the asymmetric parameter against the single scattering albedo at 440 nm. This plot



aims to elucidate the clusters distinctions related to particles absorption efficiency and the
asymmetry between hemispherical forward and backward scattering. Aerosol regimes
dominated by coarse particles tend to exhibit more significant forward scattering and,
consequently, higher asymmetry parameter values. In contrast, lower asymmetry parameter
values are expected in fine mode regimes (Eck et al., 1999, Dubovik et al., 2002). This pattern
is evident in the graphic; clusters C0 and C1 present higher asymmetry parameter values, It
is also possible to identify the distinction between the non-dust regimes C2, C3 and C4. C2
presents the lowest asymmetry parameter values while it is the most absorbing of the
clusters, according to its single scattering albedo values. Small and highly absorbing particles
are commonly associated with urban pollution or fresh smoke plumes from biomass burning
(Dubovik et al., Omar et al., 2005, Levy et al. 2010, Martins et al. 2009).  The C3 cluster differs
significantly from C2 by presenting higher asymmetry parameter values, an indication of a
shift to larger particle sizes. C3 has higher single-scattering albedo values, indicating a less
absorbing aerosol regime. SSA alone did not help to differentiate the two clusters dominated
by coarse mode particles (C0 and C1). C0 asymmetry parameter values tend to be lower than
those of C1, suggesting that the former could be a dusty mixture not as close to a pure dust
scenario as C1. The traditional plot of Lidar Ratio (LR) against Linear Depolarization Ratio
(LDR) (Kanitz et al. 2013, Illingworth et al., 2015) confirms this hypothesis (Fig. 3d). Pure
dust regimes of aerosol, due to its high level of non-spherical particles, produce higher LDR
(Groß et al., 2011). The C1 cluster presents higher values of LDR than C0, indicating that C1
is closer to a pure dust regime. The C0, while a dust regime, is likely to represent a mixed
scenario given its LDR values consistent with dust and smoke mixing (Kanitz et al. 2013).
LDR values below 15%, which is the case of the clusters C2, C3 and C4, are typically associated
with fresh/aged smoke, urban-industrial pollution, and marine particles scenarios. The
analysis of the real part versus the imaginary part of the complex refractive index (Fig. 3c)
emphasizes C2 as the aerosol regime with the largest absorption and highlights that the real
part of the complex refractive index is the main aspect differentiating C3 and C4.






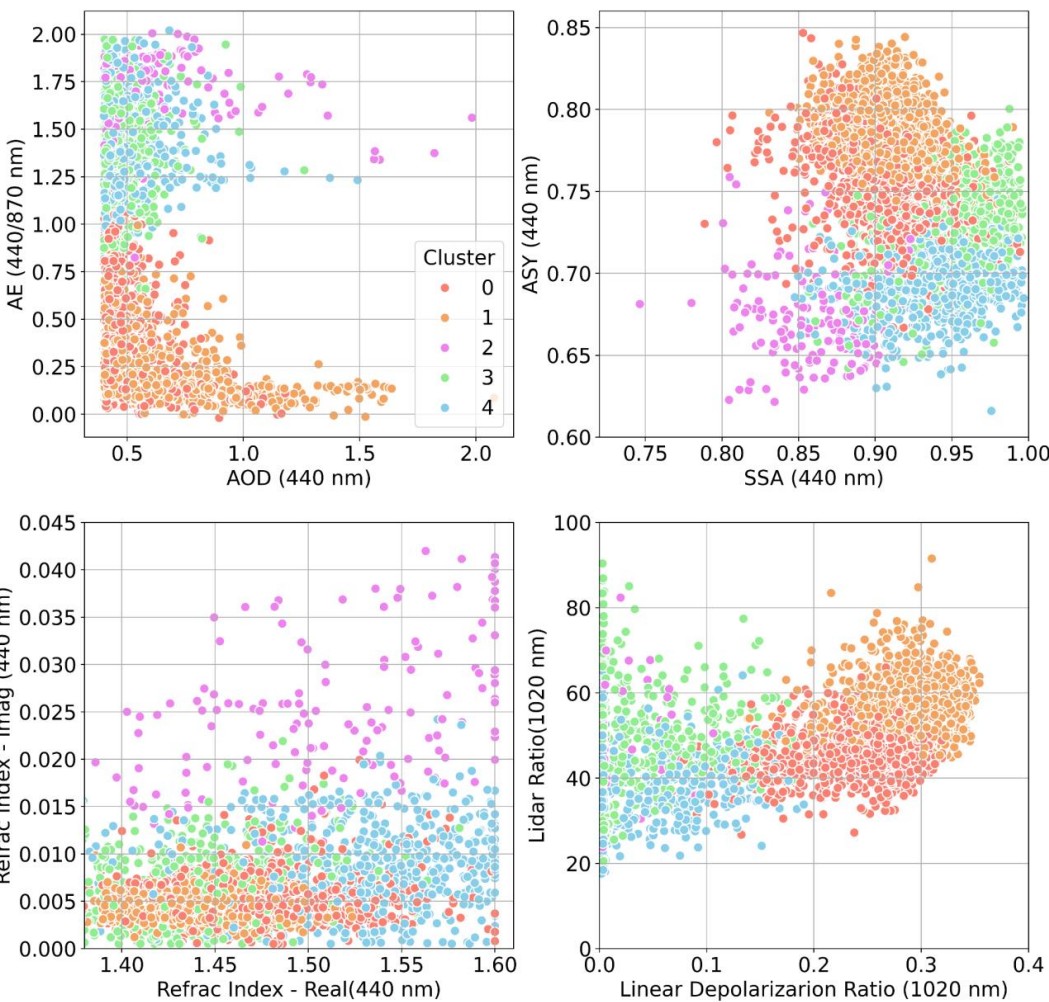


***Figure 3:*** *Scatterplot of the clusters elements as function of different parameters: a) Extinction Angstrom Exponent (AE) as function of Aerosol Optical Depth (AOD) at 440 nm; b) Asymmetry Parameter (ASY) as function of Single Scattering Albedo (SSA) at 440 nm; c) Lidar Ratio as a function of Linear Depolarization Ratio at 1020 nm; d) Refractive index at 440 nm: Imaginary part as function of Real part.*

**Figure 4** and **5** present the clusters average for selected features: size distribution, complex refractive index, single scattering albedo, and asymmetry parameter. A more detailed summary of the mean behavior of the clusters is presented in **Table 3**. The average size distribution of the clusters confirms that aerosol regimes affecting the Iberian Peninsula vary between two scenarios dominated by coarse mode (C0, C1), named here as dust regimes, and three scenarios when coarse mode is not dominant, here considered as non-dust regimes. There are differences between the dust regimes: C1 is associated with a higher coarse particle





loading than C0.  Among the non-dust regimes (C2, C3 and C4), the main difference is seen
between C2 and the two others. C2 is characterized by larger fine particles loading. Between
C3 and C4, one can observe a larger radius spread for C3 regarding the contribution of the
fine mode.

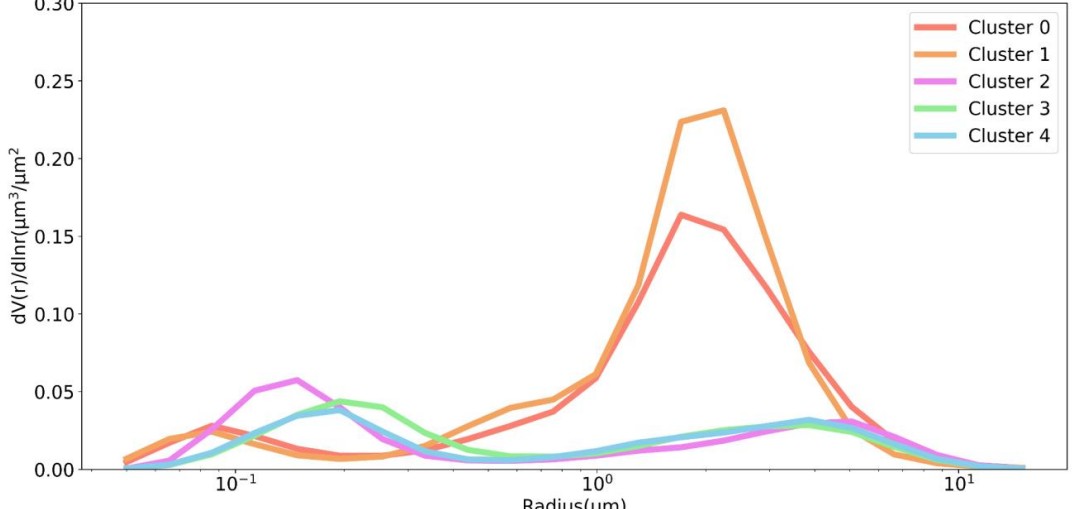

***Figure 4:*** *Clusters mean volume particle size distribution as a function of radius. The numeric*
*values of each cluster size distribution can be found in Table S2 in the supplement.*

Clusters C2 and C4 have close values for the real part of the refractive index but the former
C2 has a much larger imaginary part, justifying its lowest SSA (**Figure 5**). The C2 strong
absorption combined with its smaller particles suggest that it is likely associated with fresh
smoke (Reid e tal., 1998, Reid et al., 2005). The average of the real part of the complex
refractive index corroborates the difference between the C3 and C4 aerosol regimes.
According to Moise et al. (2015), a variation as such observed between C3 and C4 (1.4 to 1.5)
could produce an increment of 12 % in estimating the direct aerosol radiative forcing over
the solar spectrum wavelength range. Zhao et al. (2019) also showed that the direct aerosol
radiative forcing is estimated to vary by 40 % when the real part of the complex index values
varies between 1.36 and 1.56.  The reasons for the differences observed between the real
parts of C3 and C4 remain unclear. However, the spatial distribution of the clusters (see Fig.
6) indicates that C3 is more prevalent in the eastern region of the Iberian Peninsula, which is
the wettest area and more exposed to air masses from the Mediterranean and Eastern
Europe. Additionally, the low values of the real part of the complex refractive index for C3
align with aerosol regimes that have a strong contribution from sulfate particles. The spectral
dependency of the single scattering albedo corroborates our attribution of the C0 and C1 to
a dust regime. Dust particles are characterized by strong absorption in the UV spectrum
(Dubovik et al., 2002), which decreases as the wavelength increases, a feature present in both



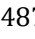



C0 and C1. Also, consistent with dust-dominated regimes, C0 and C1 have the largest mean
asymmetry parameter at all wavelengths.

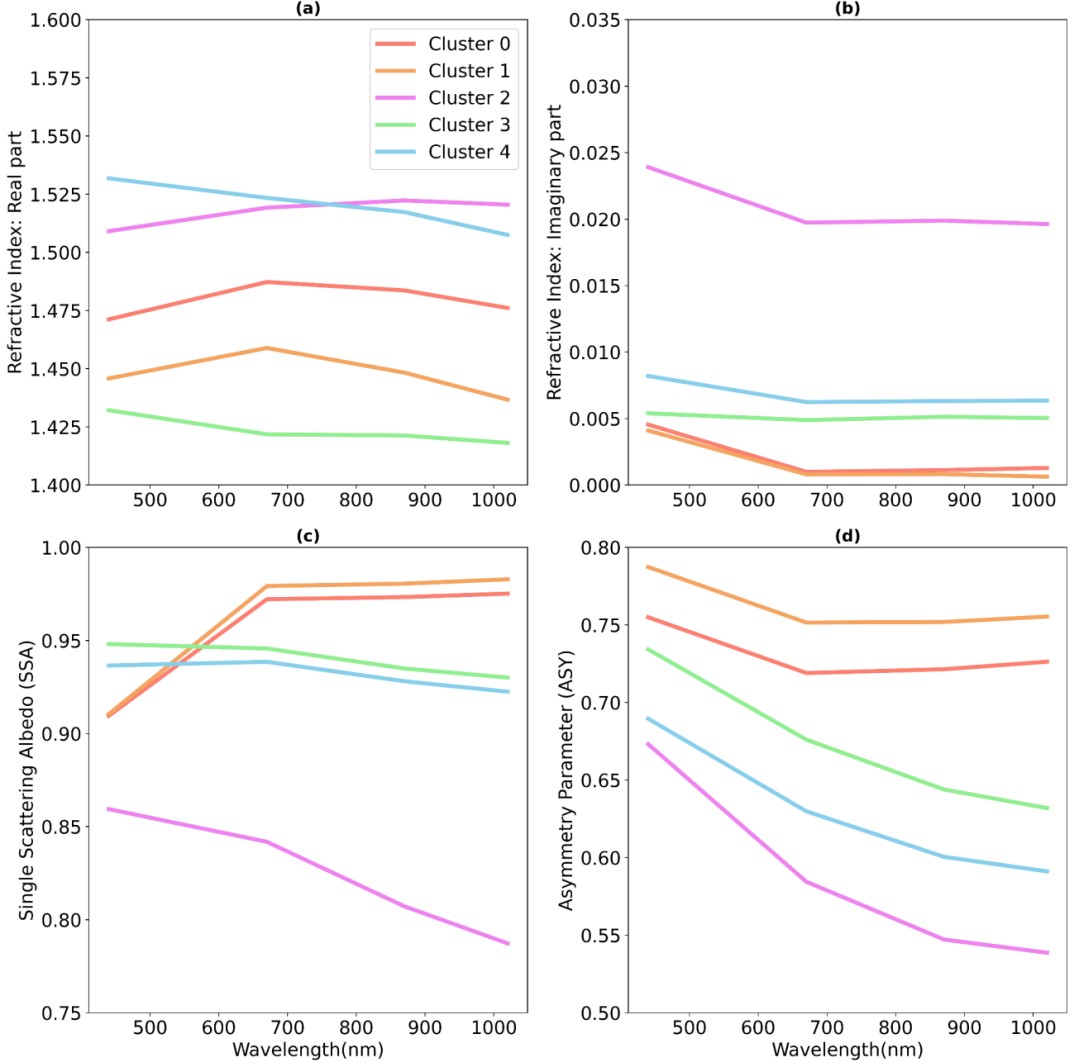


**Figure 5:** *Clusters average of complex refractive index, (a) Real and (b) Imaginary parts, (c)*
*single scattering albedo and (d) asymmetry parameter.*


The analysis above and the summary provided by **Table 3** provide several specific
characteristics that help us to contextualize the clusters. To enhance this understanding, we
add the spatial (**Figure 6**) and seasonal (**Figure 7**) distribution of the clusters into our
analysis.  C0 and C1 aerosols regimes are dominated by dust, where C1 is the closest regime



to what we could call pure dust scenario. Both aerosol regimes, C0 and C1, affect practically
the entire Peninsula (**Figure 6**) and all year round, but it is more frequent in the southern
part of the Peninsula, an expected feature considering that the dust particles are mainly
transported from North Africa (Cachorro et al., 2016, Gómez-Amo et al., 2017). The C2 cluster
is the most absorbing regime, and is characterized by the smallest fine mode particles. Our
hypothesis is that C2 is associated with fresh smoke. Its spatial distribution (**Figure 6**) with
more frequent occurrence along the belt spanning from Evora, in Portugal, to Caceres, in
Spain, a region known for high recurrence of biomass burning, reinforces our hypothesis.
Additionally, the seasonal distribution of C2 in this region coincides with the peak of the
biomass burning season.  C3 aerosol regimes also occur over all AERONET sites during all
seasons, but it is dominant in the eastern and northeastern portions of the Iberian Peninsula.
Among non-dust regimes, its unique feature is its very low real part of the refractive index.
C4, as C3, is weakly absorbing according to their single scattering albedo. However, it is
present across the entire Peninsula, but its occurrence increases in the central and in the
northern portions, which are more prone to biomass burning. An important feature of C4 is
that its occurrence increases during the summer and beginning of autumn (**Figure 7**) in the
central region of the Iberian Peninsula, from Évora (Portugal) to Madrid (Spain), when the
region's biomass burning season is going on. These aspects led us to hypothesize that C4 is
an aerosol regime under strong influence of smoke aerosol particles.

***Table 3:*** *Summary of the clusters based on the average of optical and microphysical properties.*
*A detailed description of the clusters can be found in Tables S1 and S2 in the supplement.*

| Properties | Cluster0 (Polluted dust) | Cluster1 (Pure dust) | Cluster2 (Strongly absorbing smoke) | Cluster3 (Urban-Industrial Pollution) | Cluster4 (Moderately absorbing smoke) |
|---|---|---|---|---|---|
| Number of records | 1308 | 1665 | 153 | 660 | 604 |
| Percentage (%) | 29.76 | 37.88 | 3.48 | 15.01 | 13.74 |
| Ref_Idx_Real ( 440 nm) | 1.47(0.04) | 1.44(0.03) | 1.51(0.07) | 1.43(0.06) | 1.52(0.05) |
| Ref_Idx_Img ( 440 nm) | 0.005(0.002) | 0.004(0.001) | 0.025(0.009) | 0.006(0.004) | 0.009(0.004) |
| VMR-F | 0.14(0.03) | 0.14(0.03) | 0.16(0.02) | 0.21(0.04) | 0.18(0.04) |
| STD - F | 0.61(0.09) | 0.67(0.07) | 0.42(0.06) | 0.47(0.06) | 0.41(0.05) |
| REff-F | 0.12(0.02) | 0.12(0.02) | 0.14(0.02) | 0.18(0.03) | 0.17(0.03) |
| REff-C | 1.68(0.16) | 1.61(0.13) | 2.44(0.43) | 2.31(0.38) | 2.25(0.49) |
| VMR-C | 2.02(0.23) | 1.88(0.17) | 3.10(0.45) | 2.82(0.42) | 2.82(0.57) |
| STD-C | 0.60(0.52) | 0.54(0.04) | 0.68(0.06) | 0.63(0.05) | 0.67(0.05) |
| AOD (440 nm) | 0.50(0.11) | 0.58(0.21) | 0.64(0.29) | 0.48(0.09) | 0.51(0.13) |
| SSA (440 nm) | 0.91(0.03) | 0.91(0.02) | 0.86(0.03) | 0.95(0.03) | 0.94(0.03) |
| ASY (440 nm) | 0.76(0.02) | 0.79(0.19) | 0.67(0.03) | 0.73(0.03) | 0.69(0.02) |
| AE(440/870 nm) | 0.40(0.25) | 0.24(0.14) | 1.67(0.20) | 1.43(0.26) | 1.47(0.25) |
| LR(1020 nm) | 64(9) | 70(8) | 89(16) | 77(17) | 61(15) |
| LDPR(440 nm) | 0.17(0.04) | 0.21(0.04) | 0.01(0.03) | 0.03(0.04) | 0.03(0.05) |








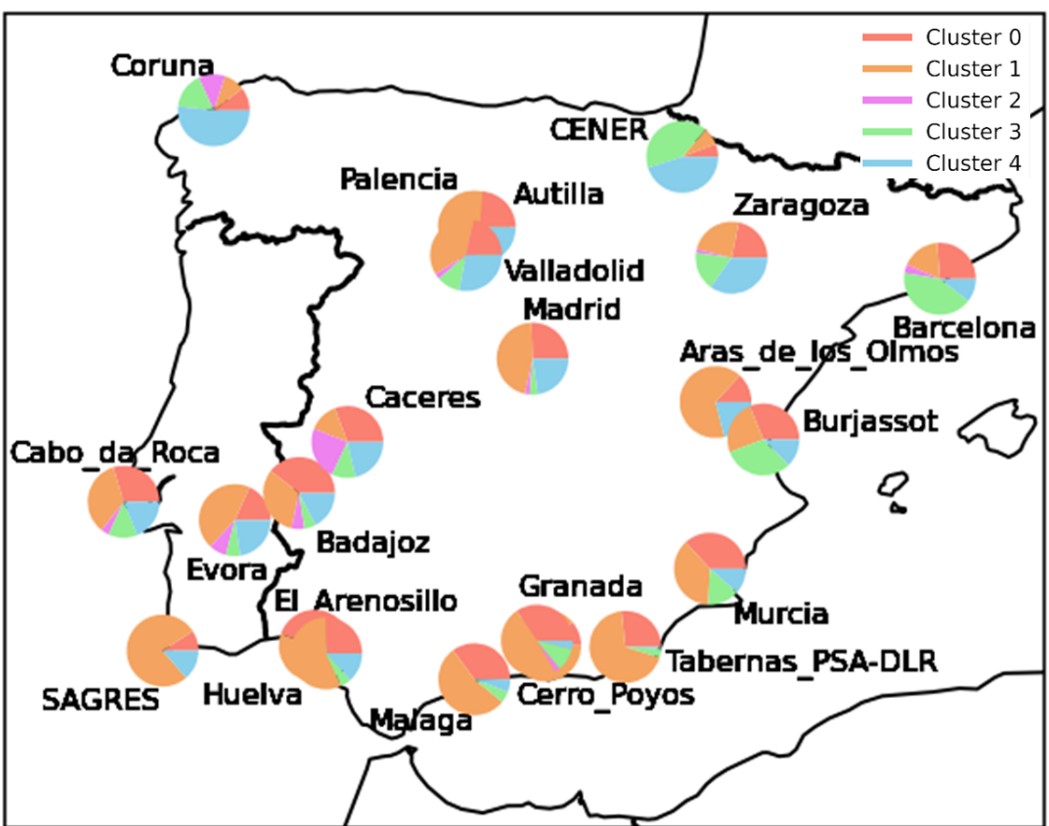


***Figure 6: Proportions*** *of the occurrence of the clusters of aerosol regimes at the AERONET*
*sites across the Iberian Peninsula.*


**Figures 7** and **8** provide a perspective view on the seasonal occurrence of each cluster based
on sites that represent different regions of the Iberian Peninsula.

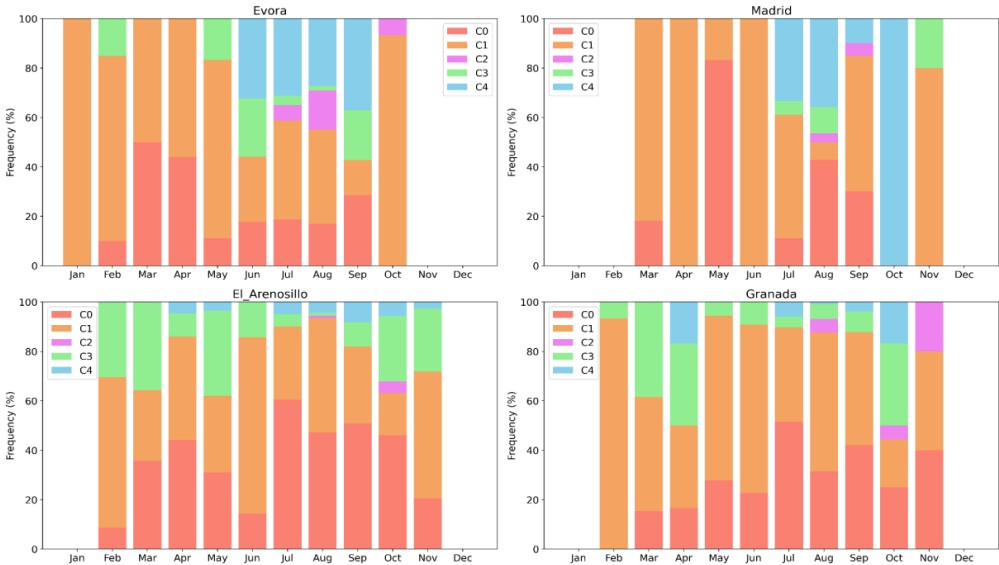


**Figure 7**: *Clusters relative monthly occurrence over the AERONET sites representatives of the Iberian Peninsula western lowlands (Evora), highlands plateau (Madrid) and southeast lowlands (El Arenosillo, Granada).*

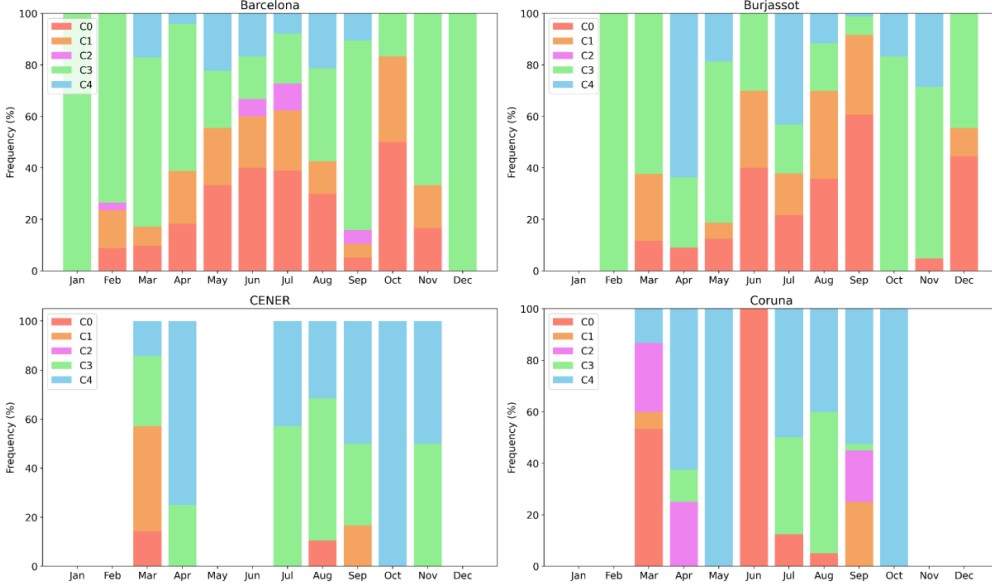


**Figure 8:** *Clusters relative monthly occurrence over the AERONET sites representatives of the following Iberian Peninsula regions: Eastern Coast (Barcelona, Burjassot) and Northern (Coruna, CENER).*





**3.2 Random Forest Classifier: Performance and Optical models spatial dynamic**

The Random Forest training of MERRA-2 aerosol-type column mass density as predictors of aerosol optical regime covered 70% of the AERONET sky inversions used in this study, combining datasets from all sites. The testing dataset, constituted by the remaining 30%, was used to evaluate the model performance. The best parameters obtained from the optimization using RandomizedSearchCV were the number of decision trees of 477 (n_estimators = 477) and maximum depth of trees of 19 (max_depth=19). There are several metrics for accessing machine learning performance. **Figure 9** presents the one used in this study, the Normalized Confusion Matrix (NCM), which expresses the percentage of correct and incorrect predictions (where the classifier got confused). In the matrix, the rows represent the true labels, and the columns represent the predicted ones. The values along the diagonal indicate the percentage of times where the predicted matches the true label. The other cells reflect instances where the classifier mislabeled an observation; the column tells us what the classifier predicted, and the row tells us the correct label.

For all clusters, the classifier's correct predictions surpassed the incorrect predictions, with a maximum frequency of correct prediction close to 80% obtained for C1. The minimum percentage of correct prediction, about 60%, was obtained for C2, the highest absorbing cluster. Regarding dust regime clusters, despite the struggle to predict C0, it is possible to see that, in this case, the classifier´s main confusion is with the C1, which is also a cluster related to an aerosol scenario dominated by coarse mode particles (dust regime), as C0. Therefore, this is a somehow expected confusion, which would not introduce a substantial error in the radiative effect calculations. Rarely does the classifier take either C0 or C1 as C2, C3, and C4, a case where substantial error in the radiative effect would be expected. By combining C0 and C1 results in the NCM, the percentage of correct prediction achieved by the classifier indicating dust regime is higher than 95%. Similarly, rarely the classifier takes C3 and C4 as C0, C1 and C2. Given that C3 and C4 are also close in terms of their optical properties, especially concerning absorption, some degree of confusion among them is expected. Nevertheless, these aspects of the confusion matrix among close clusters are important to identify where the model needs extra training. C2, the less frequent and the one representing the most absorbing aerosol regime over the Iberian Peninsula is rarely mislabeled as C0 or C1, but often mislabeled as C3 or C4. Still, the score percentage is around 60%.




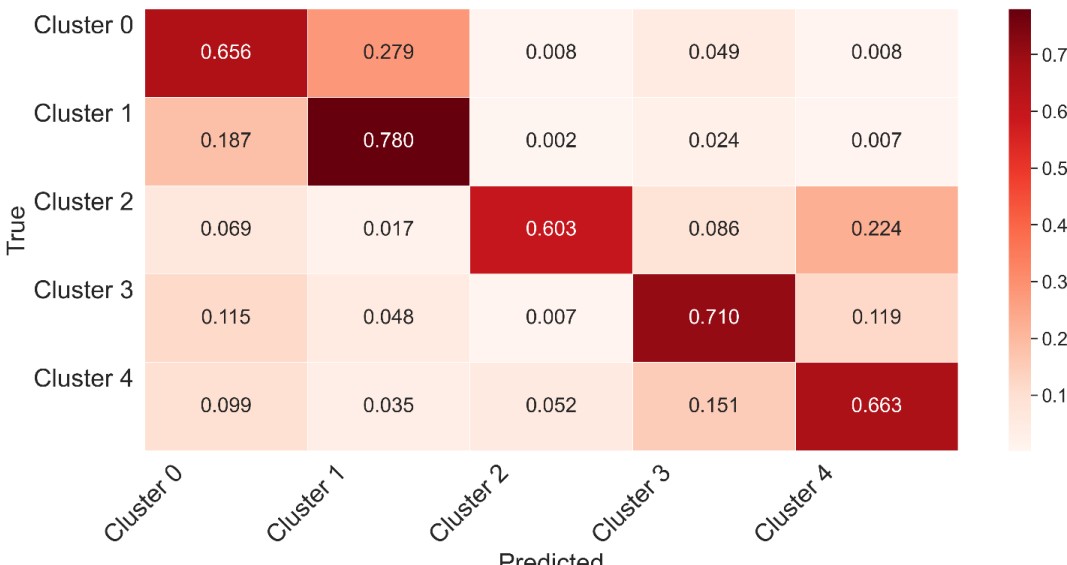


*Figure 9: Normalized confusion matrix of the Random Forest classifier applied to the prediction of the clusters that describe the typical aerosol optical regime based on MERRA-2 aerosol components column mass density.*

To provide further insight into the model performance, we also examined other metrics
commonly used to evaluate Random Forest training: Accuracy, Precision, Recall, and F1 score
(**Table 4**). The trained model achieved a general accuracy of 70 %, meaning it correctly
predicted the aerosol regime in three out of four cases. For all clusters, all metrics adopted
were higher than 0.60, with precision and recall values exceeding 0.75 in some cases. The
precision metric indicates how often the positive predictions are correct. The model
precision varied within the specific optical regimes (ex., non-dust) and among optical
regimes (dust, non-dust). It showed higher precision in identifying C1 than C0, the two dust-
regimes. Among the non-dust regime clusters, the highest precision obtained was related to
the prediction of C2, suggesting a lower likelihood of false positive for this class given its
strong absorption nature, mislabeling this aerosol regime would translate in high cost due to
significant radiative error; therefore, its highest precision is a promising outcome.

**Table 4**. Performance metrics values of the trained model prediction of aerosol optical
regime based on aerosol-type column mass density.

| Clusters | Precision | Recall | F1-Score | Support (N) |
|---|---|---|---|---|
| 0 | 0.62 | 0.62 | 0.62 | 394 |
| 1 | 0.68 | 0.70 | 0.69 | 452 |





| 2 | 0.62 | 0.60 | 0.61 | 62 |
| 3 | 0.76 | 0.73 | 0.74 | 251 |
| 4 | 0.68 | 0.69 | 0.69 | 185 |


**Figure 10** illustrates the relative importance of the predictor variables, highlighting the
influence of each aerosol-type column mass density on the model's decision-making. The
results indicate that the presence of dust and organic carbon over the Iberian Peninsula is
the primary factor affecting the aerosol optical properties in this region. This finding aligns
with actual conditions, as the transport of Saharan dust to the peninsula and biomass burning
are the two main sources driving the variability of aerosol optical properties in the area.
Interestingly, black carbon column mass density did not rank among the top predictors.
Despite the expectation that black carbon might serve as a significant indicator of the aerosol
optical regime due to its association with smoke-influenced aerosols, there is considerable
uncertainty in black carbon simulations in atmospheric chemistry models, including
reanalyzes such as MERRA-2. This uncertainty may hinder its effectiveness in predicting the
aerosol regime observed at AERONET monitoring sites.

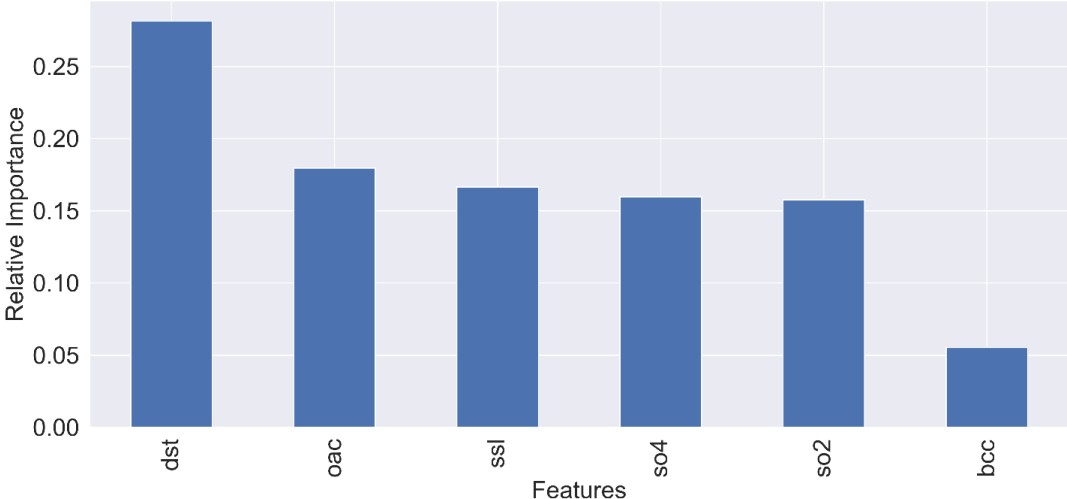


***Figure 10****: Relative importance of the predictor variables, i. e. the degree of influence of each*
*aerosol-type column mass density on the model decision-making.*





**3.3 Application: Case studies**

From the testing dataset, we selected some case studies that significantly impacted local populations, garnered media attention, and represented different aerosol scenarios in the Iberian Peninsula. This selection provides a visual (qualitative) demonstration of the model´s predicting capability (**Table 5**).

**Table 5:** List of case studies of aerosols high loading events over Iberian Peninsula selected to highlight as examples of the classifier trained model application.

| **Case study** | **Date** | **Nature (Reference link)** |
|:---:|:---:|:---:|
| #01 | June 27, 2023 | Smoke[1] |
| #02 | October 16, 2017 | Dust and Smoke[2] |
| #03 | August 11, 2016 | Smoke[3] |
| #04 | March 17, 2022 | Dust[4] |

1-https://earthobservatory.nasa.gov/images/151507/canadian-smoke-reaches-europe

2-https://atmosphere.copernicus.eu/saharan-dust-and-smoke-over-france-and-uk

3-https://earthobservatory.nasa.gov/images/88552/fires-rage-in-portugal

4- https://earthobservatory.nasa.gov/images/149588/an-atmospheric-river-of-dust

We set our trained model to prescribe the spatial distribution of aerosol optical regimes (clusters) that best fit various scenarios based on MERRA-2 aerosol-type column mass density. The results for all cases studied are presented in **Figure 11**. Since AERONET sky inversion products only provide a complete characterization of aerosol microphysical (size distribution plus complex refractive index) and optical properties (Asymmetry parameter and Single Scattering Albedo) for conditions of AOD at 440 nm exceeding 0.4, we will only discuss the optical regime prescriptions for areas where AOD was above this threshold.

 For our analysis, we used the MERRA-2 AOD field as a reference.

 Case#01 occurred from June 1 to 25, 2023, coinciding with large-scale wildfire events in Quebec, Canada.  A substantial portion of smoke from these wildfires crossed the Atlantic Ocean and reached Western Europe, especially the Iberian Peninsula, resulting in darkened skies in the affected countries.  Our trained model predicted that the most suitable aerosol optical regime for the areas impacted by the smoke (Portugal, Western, and Northern Spain) is C4, which corroborates our previous discussion associating the C4 optical regime to regional smoke.

Case#02 features an emblematic event on October 16, 2017, marked by simultaneous massive wildfire in central and northern Portugal and a strong dust transport from North





Africa via the south of Portugal.  The corridor connecting the smoke and dust produced a
strong northward transport affecting the United Kingdom, influenced by the synoptic
conditions associated with the ex-hurricane Ophelia, located just north of the Iberian
Peninsula (Osborne et al., 2019). The optical regime prescription identified the C4 cluster as
the appropriate regime from central Portugal northward to the UK. Meanwhile, the area
affected by dust, spanning from North of Africa to southern and central Portugal, was
characterized by a mix of C0 and C1, the clusters associated with dust regimes. As the dust
plume arrived in Portugal, the model indicated a gradual transition from C1, indicative of
pure dust, to C0, which represents conditions of dust mixed with smoke (Gómez-Amo et al.,
2017). The random distribution of C2 within the larger C4 regions likely reflects the model´s
response to the specific conditions dictated by the aerosol-type column mass densities. This
could suggest patches of high-absorbing aerosol-type within a less-absorbing large-scale
smoke plume, although there is insufficient evidence to draw definitive conclusions.
Case#03, dated August 16, 2016, involved strong wildfire emissions in northern Portugal.
Most of the smoke was transported toward the Atlantic Ocean, while the remainder of the
peninsula experienced low aerosol loading conditions. Consistent with smoke aerosol
scenarios, the model prescribed the C4 optical regime.
Case#04 pertains to an extreme Saharan dust transport that affected most of the Iberian
Peninsula on March 15-17, 2022. During this event, the 24-hour average concentration of
PM2.5 reached as high as $700\,\mu g\,m^{-3}$ in parts of Spain (Rodriguez and López-Darias, 2024).
The pollution episode was dominated by dust, and indeed, the model prescribed the optical
regime C1, which indicates the pure dust conditions for most of the Iberian Peninsula. This
demonstrates our approach´s capability to differentiate specific scenarios within dust
regimes.  For non-dust regimes regarding C2, a highly absorbing regime, we would not expect
to see widespread prescriptions, as we hypothesize that it is associated with fresh, high-
absorbing pollution plumes. **Figure 6**, depicting the occurrence of each cluster across the
Iberian Peninsula, corroborates our hypothesis by indicating that the C2 regime is mainly
present in specific areas where aerosol loading increases are primarily attributed to biomass
burning, such as the western lowlands of Iberian Peninsula (Evora, Badajoz, and Caceres) and
in the Galicia region (Coruna). The C3 optical regime was not linked to large-scale dust
transport or smoke plumes across the Iberian Peninsula, suggesting it might be associated
with high levels of local or regional pollution. **Figure 6** shows that the C3 regime is commonly
observed throughout the year in the eastern portion of the Iberian Peninsula. The results of
these case studies, combined with performance evaluations, highlight the capability and
potential of this machine-learning approach, which uses clustering and random forest
classification to prescribe optical models from aerosol-type columnar mass density to
calculate aerosol particles' direct radiative effect in atmospheric models. By constraining
modelling with observational data, we can help mitigate the known uncertainties related to
aerosol direct radiative forcing. Additionally, our method's straightforwardness and lower
computational cost favor operational modeling when infrastructure is limited.



**Figure 11**: *Case studies of distinct aerosol scenarios over the Iberian Peninsula selected to test our machine-learning based approach to predict the best optical property regime: (a) Case#01 on June 27, 2023; (b) Case#02 on October 16, 2017; (C) Case#03 on August 11, 2016; (d) Case#04 on March 15, 2022. On the left side, MODIS/NASA True color satellite images (https://wvs.earthdata.nasa.gov); and on the right the cluster spatial distribution prescribed by the model.*




**Figure 12** shows the single scattering albedo at 550 nm field, comparing the current approach and MERRA-2 reanalysis results. The MERRA-2 columnar total SSA was calculated based on the ratio of total scattering aerosol optical depth to total extinction aerosol optical depth. For smoke scenarios on June 27, 2023, MERRA-2 indicated a more absorbing optical regime (SSA at 550 nm ~ 0.86 - 0.90) compared to the current approach (SSA at 550 nm ~0.95). On this day, the average SSA at 550 nm over the AERONET site in Coruna City, which was directly affected by Canadian smoke, exceeded 0.95. A similar trend was observed for the dust scenarios. For example, on March 17, 2022, the current approach prescribed a less absorbing optical regime (SSA at 550 nm ~0.94 - 0.95) compared to MERRA-2, which reported a SSA at 550 nm of roughly 0.92 - 0.94. The analysis of SSA at 550 nm over AERONET sites affected by the dust event surpassed 0.94. While these cases highlight differences between the prescriptions based on the clusters and MERRA-2 results, they are only sufficient to warrant further investigation. To gain a statistical perspective on whether the findings from these case studies are isolated incidents or indicative of a trend, we compare a much larger sample of MERRA-2 SSA at 550 nm across various AERONET sites in the Iberian Peninsula using the clusters approach. We focused only on MERRA-2 aerosol scenarios for AOD at 550 nm larger than 0.3 and conducted the comparison segmented by the optical regimes defined by the clusters.

**Figure 13** shows the count distribution of MERRA-2 SSA at 550 nm for the aerosol regimes represented by the clusters C0, C1, C3 and C4, as classified by the random forest classifier we developed. Each cluster's SSA at 550 nm histogram was randomly simulated following a Gaussian distribution considering the cluster mean and standard deviation. A similar analysis was conducted for the Angstrom Exponent (**Figure 14**). In addition to the absorption, we evaluated aspects of mean particle size behaviors. Based on **Figure 13**, we found that, on average, our aerosol optical regime prescription based on the clusters (AERONET) is less absorbing than MERRA-2 for aerosol regimes C0, C1, C3 and C4. More significant differences are observed for C1, C3 and C4. Cluster C1 corresponds to a dust scenario closer to pure dust, while C4 is dominated by smoke. Regarding the particle size indicator (AE), it was observed that MERRA-2 has a lower contribution of coarse particles in the dust regimes compared to the cluster-based prescriptions (**Figure 14 a, b**). This finding aligns with Adebiyi et al. (2023), which noted that climate models tend to underestimate large dust particles, mainly when representing North African dust plumes. Conversely, for the non-dust regimes (C3, C4), MERRA-2 shows a larger relative contribution of coarse particles than the clusters-based prescription (**Figure 14 c, d**). **Figure 15** shows the results for C2. For this specific regime, on average, prescription based on the cluster (AERONET) is more absorbing than MERRA-2, opposite to the findings of the other clusters. Regarding AE, under the C2 regime, MERRA-2's mean AE is lower than that prescribed from the cluster, suggesting a lower relative contribution of fine mode in the reanalysis simulations. This is like the findings related to the two other fine mode dominant regimes (C3 and C4).

As demonstrated by the SSA and AE distributions (Figures 13, 14, 15) and evaluated from Table 1 and 4, the model can also predict the occurrence of the minority cluster C2 (3–4 percent of samples). The model preserves the physical distribution characteristics of less frequent aerosol regimes while capturing its features without the need for explicit class imbalance treatment. With C2's highly absorbing and dominant fine mode conditions





reflected in both SSA and AE predictions, the distributions across clusters demonstrate
agreement between expected and observed distribution values.


**Figure 12**: *Single Scattering Albedo (SSA) prescription based on the current study approach*
*(left) and that simulated by MERRA-2 (right) for the selected case studies of Table 2: (a)*
*Case#01 on June 27, 2023; (b) Case#02 on* October 16, 2017; *(C) Case#03 on August 11, 2016;*
*(d) Case#04 on March 15, 2022.*





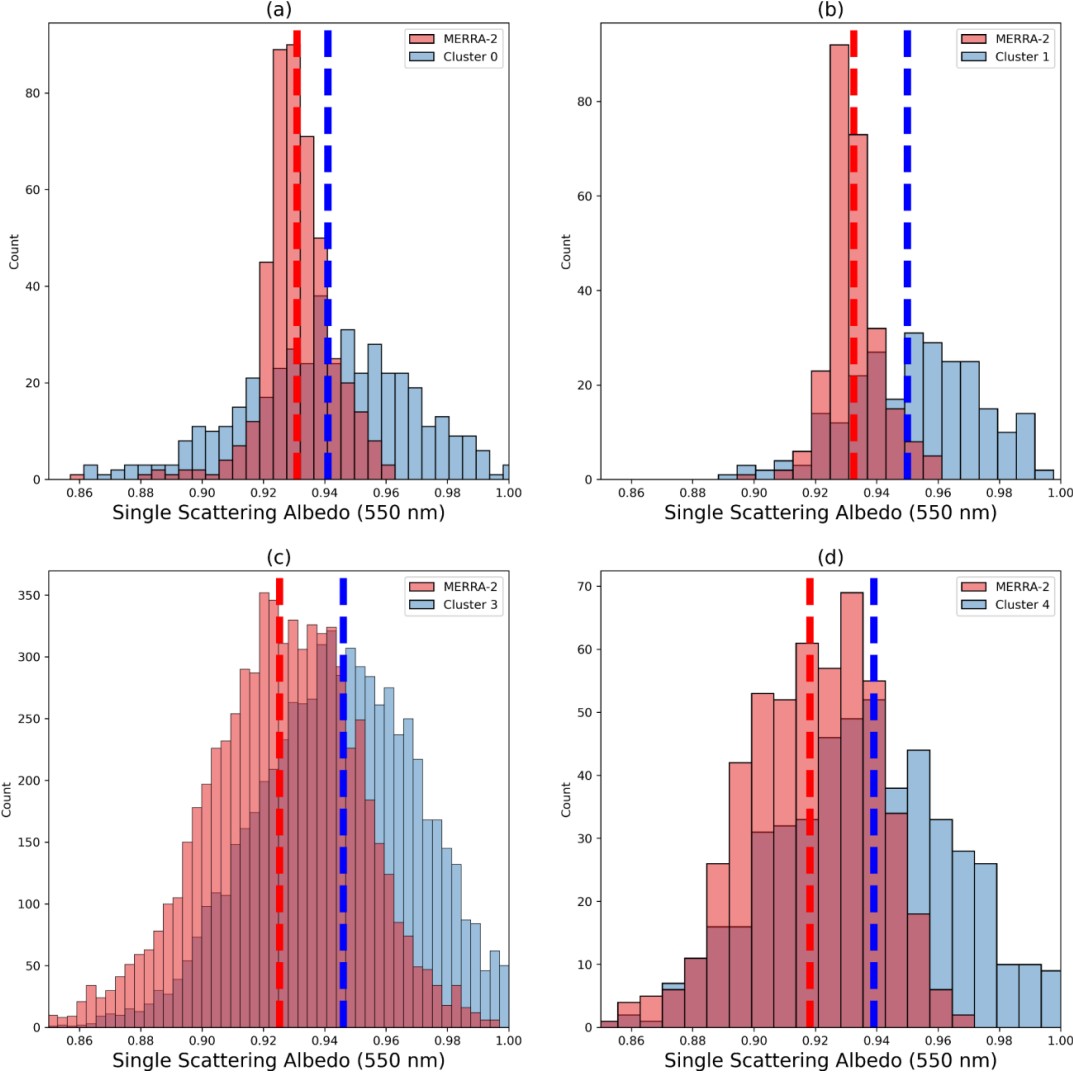

**Figure 13**: Current study prescription and MERRA-2 simulation of Single Scattering Albedo (SSA) frequency distribution as function of the optical regime (clusters): a) Cluster 0; b) Cluster 1; c) Cluster 3; e) Cluster 4.



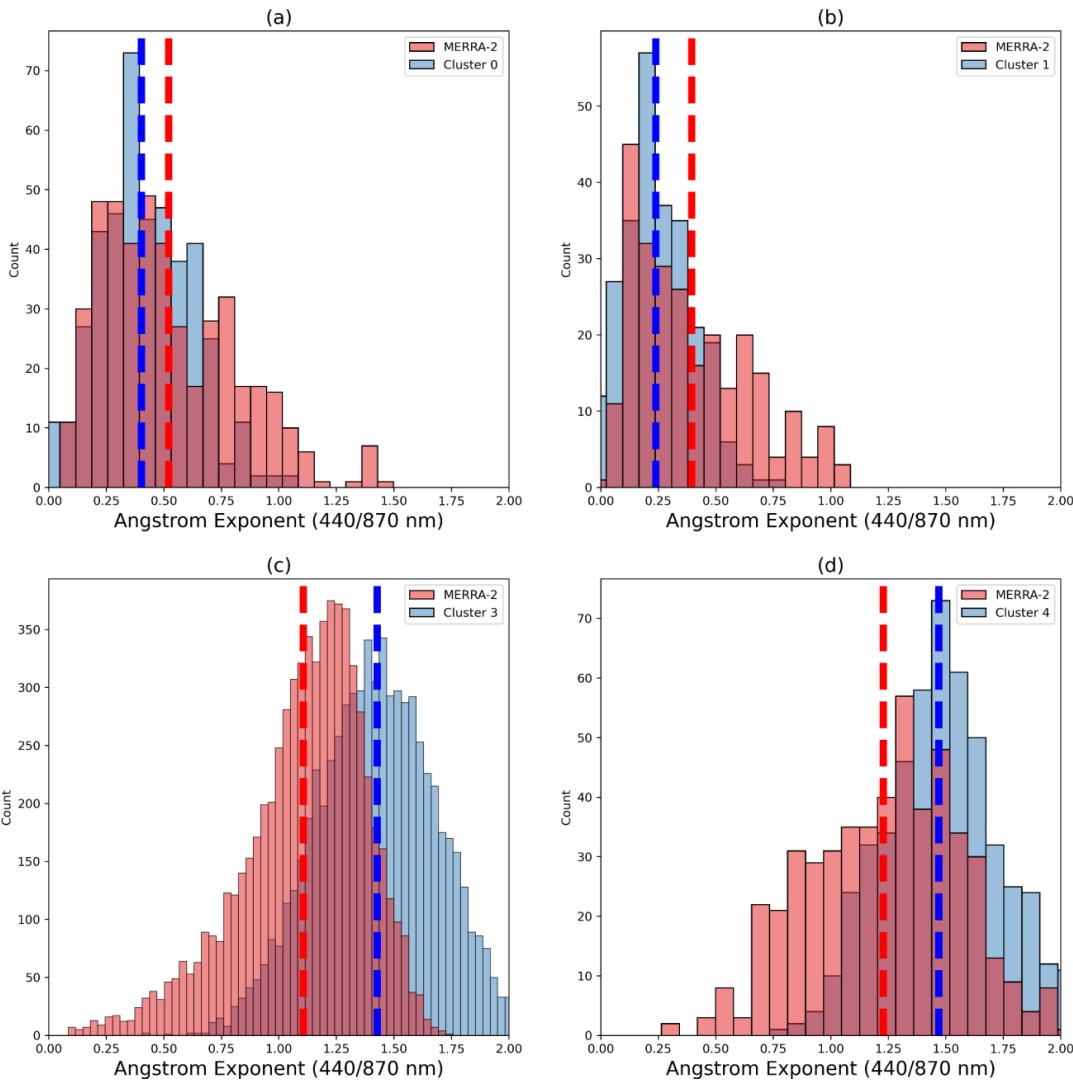


**Figure 14**: Current study prescription and MERRA-2 simulation of Angstrom Exponent (AE)
frequency distribution as function of the optical regime (clusters): a) Cluster 0; b) Cluster 1;
c) Cluster 3; e) Cluster 4.



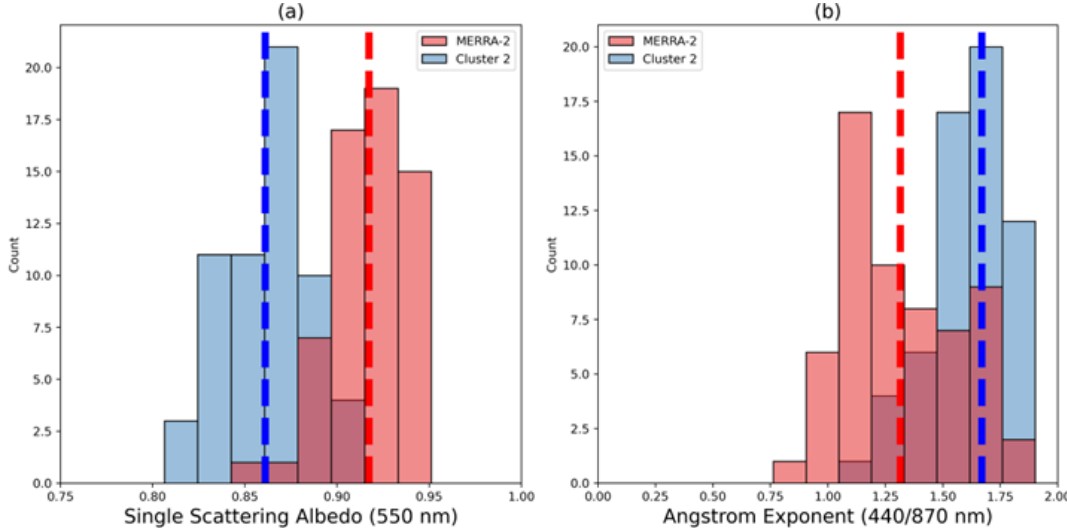



**Figure 15**: Current study prescription and MERRA-2 simulation of (a) Single Scattering
Albedo and (b) Angstrom Exponent (AE) frequency distribution for the Cluster 2 scenario.


## 4. Conclusions

This study emphasizes the importance of observational-based research to constrain the
prescription of aerosol-intensive properties in atmospheric models. We aimed to
characterize the typical aerosol intensive optical properties affecting the Iberian Peninsula
(IP) using data from the atmospheric column AERONET sky inversion products. We
employed K-means clustering to analyze historical aerosol intensive properties across all
AERONET sites that operated for at least two years and had the highest quality dataset level
(2.0) available. We identified five distinct clusters (C0, C1, C2, C3, C4) representing different
optical regimes, illustrating the predominant aerosol scenarios in the IP. The key difference
among these clusters lies in the contribution of coarse mode particles and their absorption
efficiency. Clusters C0 and C1 are dominated by coarse mode particles and classified as dust-
regimes due to their association with Saharan dust transport. In particular, the optical
properties of C1 closely resemble a pure dust scenario, while C0 indicates a more mixed
situation, which we refer to as dusty. On the other hand, clusters C2 and C4 are identified as
non-dust regimes, linked to strong and moderate absorption related to smoke plumes.
Cluster C3, also a non-dust regime, is more frequently observed in the eastern part of the IP
and differs from C4 mainly by having a much lower real part of the refractive index. After
identifying the typical aerosol regimes affecting the IP, we utilized aerosol-type columnar
mass density data (dust, organic carbon, black carbon, sea salt, and sulphates) from MERRA-
2 to predict the aerosol optical regime at each grid point using the supervised learning





methodology Random Forest. We tested the performance of the trained model under various
aerosol scenarios. The accuracy of the predictions for the aerosol optical regimes ranged
from 60% to 75%, depending on the regime, with an average accuracy of 70%. Notably, the
accuracy exceeded 90% when predicting solely dust or non-dust optical regimes.
An analysis of MERRA-2 simulations alongside this study´s AERONET cluster-based
prescriptions of optical regime indicators, such as absorption (SSA) and size (AE), reveals
that MERRA-2 is generally more absorbing for the aerosol optical regimes (C0, C1, C3 and C4)
impacting the atmosphere of the Iberian Peninsula, except for the most absorbing
regime(C2). Specifically, the reanalysis simulations indicate higher absorption under the
non-dust regimes C3 and C4. When examining the relative contributions of fine and coarse
modes, the cluster-based prescription indicates a larger average contribution of coarse
particles than the MERRA-2 under dust-regimes (C0, C1). Conversely, for the non-dust
regimes (C2, C3, C4), MERRA-2 shows a lower relative contribution from the fine mode
compared to the clusters-based prescription.
Our findings contribute to enhancing the understanding of the dynamic aerosol optical
properties over the Iberian Peninsula and highlight the potential of machine-learning
approaches to improve the representation of aerosol radiative forcing in atmospheric
models. Many atmospheric modelling systems are not designed to simulate aerosol-intensive
microphysical and optical properties in real time. Additionally, computational cost remains
a common limitation worldwide. Our approach integrates AERONET-derived intensive
properties based on climatological optical regimes to refine the model, coupled with
predicted aerosol-type columnar mass density. This integration can help reduce regional
uncertainty in the simulation of aerosol radiative forcing.

## Competing interests

The authors declare that they have no conflict of interest.

## Acknowledgements and financial support

The authors acknowledge the financial support of FCT—Science and Technology Portuguese
Foundation, which funded the project FIRESMOKE (http://doi.org/10.54499/PTDC/CTA-
MET/3392/2020) through national funds. Thanks are also owed to the financial support
given to CESAM by FCT (UID Centro de Estudos do Ambiente e Mar (CESAM) +
LA/P/0094/2020) through national funds. We also acknowledge the financial support of
CNPq - National Council for Scientific and Technological Development (CNPq) through the
funding processes CNPq Nº 441851/2023-1 and CNPq Nº 172486/2023-8. Author HFCV also
thanks to the CNPq grant No 315349/2023-9. We thank AERONET and MERRA-2 PIs and
teams for their effort in establishing and maintaining the sites and the reanalysis
development used in this study. We acknowledge the use of imagery from the Worldview
Snapshots application (https://wvs.earthdata.nasa.gov), part of the Earth Observing System
Data and Information System (EOSDIS).



**Author contributions**
NR, KL and PT designed and performed the research, analyzed the data, and wrote the first
version of the paper. MY, SF, LF, OM, HFCV contributed to writing, discussion, review and
editing. ICM and AIM conceptualization and coordination of the Project FIRESMOKE,
discussion, review and editing.
**Code and data availability.**
All the datasets (AERONET and MERRA-2) used in this study are publicly available and were
downloaded from their respective websites (https://aeronet.gsfc.nasa.gov/; and
https://disc.gsfc.nasa.gov/datasets?project=MERRA-2). Code and dataset required to
conduct the analyses herein is available at https://doi.org/10.5281/zenodo.15178347
(Rosario, 2025).

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
