# Peer review of "Machine learning-driven characterization and prescription of aerosol optical properties for atmospheric models"

_EGUsphere, 2025_

## Author Comment (AC1)

**RC1: 'Comment on egusphere-2025-454', Anonymous Referee #1**

The authors use K-means clustering on AERONET data to identify different regimes of aerosol optical properties over the Iberian Peninsula, and subsequently train random forests to predict these regimes from aerosol column densities provided by MERRA-2. While the paper is interesting and fits the journal, I would still recommend major revisions, please refer to comments below:

We would like to thank the Anonymous Referee #1 for the willingness to review the manuscript and for the insightful questions and suggestions, which have undoubtedly contributed to improving the clarity of our paper. While recognizing the importance of certain questions and concepts, we have chosen to further discuss them here for clarification, but omitted or summarized some of them in the manuscript since we understand that they are aspects familiar to the atmospheric aerosol modelling community. We aim to keep the article as concise as possible, given its already long extension.We hope the reviewer understands this need.

The main objective of the manuscript is to present an alternative approach for atmospheric aerosol/climate modeling communities to prescribe aerosol spectral optical properties(optical models) in climate models. Our prescription strategy incorporates a decision criterion based on the combination of mass loading of aerosol species (Dust, Sea salt, Black carbon etc.) present in each grid point air mass. This approach seeks to produce a more realistic representation of the spatial and temporal dynamics of intensive optical properties in climate models, which are needed for calculating the Aerosol Optical Depth(AOD), the key variable for simulating the Direct Radiative Effect of aerosol particles. This represents an advancement over conventional methods that rely on static areas of influence and predefined seasonality based on aerosol source emission activity (ex. Hoelzemann et al., 2009, Rosario et al., 2013, Chen et al., 2024), and also a reference to confront the approach based on prescription of individual aerosol types' optical properties and assumptions of mixing states (Collow et al., 2024).

**Major comments:**

Ln 44. "limitations in the current …. address information?" What do you mean with "address information"?

**R**: We refer to the limitation in the current global aerosol monitoring system (ground-based and satellite-based) dedicated to characterizing the spatial and temporal distribution of the spectral complex refractive index and size distribution of aerosol particles. We recognized that the expression "address information" is not clear enough to express what we meant, so we adjusted the text as follows:

*"For instance, when it comes to aerosol direct interaction with radiation, the current global aerosol monitoring system does not provide a comprehensive spatial-temporal characterization of spectral complex refractive index and size distribution of the aerosol particles, two critical information to characterize the particle absorption and scattering (Samset et al. 2018, Li et al., 2022). This lack of observational data contributes significantly to uncertainty in aerosol modeling and, therefore, on the aerosol radiative forcing.."*

Ln 48. "geographical representativity" Can you be more elaborate on what you mean here? Adebiyi et al. (2023) mention that libraries typical assume identical mineral compositions of certain aerosol species regardless of location. Is that what you mean here?

**R**: Yes, mineralogy is a good example, dust physical properties do vary from region to region depending on the soil and mineralogy composition. Additionally, other types of aerosol mixtures (urban, smoke) also differ from region to region, depending on the energy matrix and the type of biomass burned. Historically, look-up tables of optical and microphysical properties have been important to characterize and represent aerosol direct radiative effects in climate models, but their geographical limitations have also been acknowledged. So, alternative ways to capture more realistic geographical representativity of aerosol optical properties are still a matter of ongoing research. We adjusted the text as follows:

*"The difficulty of the traditional libraries of aerosol optical and microphysical properties (Shettle and Fenn, 1979; Koepke et al., 1997; Hess et al., 1998) to describe aerosols properties geographical variation, for instance soil dust mineralogy* (Adebiyi et al., 2023)*, has been central in the aerosol optical properties uncertainty debate."*

Ln 63-66. I do not fully understand the contrast between "absorption" and "relative contribution of .." implied here, are absorption problems not one of the consequences of misrepresenting the fine and coarse modes?

**R**: To a certain degree, yes, but not necessarily; some aspects drive absorption and those are not regulated by size. Soil dust, which is a dominant component of coarse mode, can be strongly absorbing, and black carbon (BC), a fine mode aerosol type, is also strongly absorbing, and the state of mixture (internal or external) of BC with other aerosol types is a critical aspect driving uncertainties in aerosol absorption in climate models. Also, one can prescribe (or predict) dust size distribution accurately and miss its absorption due to a lack of accuracy in its mineralogy composition. Regarding the misrepresentation of fine/coarse mode relative contributions, it has important implications for the scattering process.

Ln 76. Bias in what quantity?

**R**:We adjusted the text as follows, so it can be clearer

Former text that include *"bias in aerosol simulations*" was replaced for new one:

**Former text:** *"Zhong et al. (2022) used relationships from an ensemble of aerosol models and satellite observations to identify the primary source of uncertainty in aerosol modelling results. Their study pointed out the incorrect lifetimes and the underestimation of mass extinction coefficients as the most critical drivers of bias in aerosol simulations."*

**New text:** *"Zhong et al. (2022) used relationships from an ensemble of aerosol models and satellite observations to identify the primary source of uncertainty in aerosol modelling results in biomass burning regions. Their study pointed out the incorrect simulations of lifetimes and the underestimation of mass extinction coefficients as the main reasons for their difficulty in matching observed aerosol optical depth (AOD).*

Ln 107-110: This sentence is a bit ambiguous to me, do you mean that (1) dust regimes and (2) smoke regimes are the major source of differences? If so, could you rephrase?
**R**: The reviewer is correct; the sentence needs clarification. We adjusted the text to better express its goal.

*"They found larger differences between the strong and moderately absorbing aerosol regimes, namely dust and smoke regimes, when comparing global and regional clustering results. This is a consequence of the differences between China´s regional dust and smoke aerosol particles´ physical and chemical characteristics and those of global dust and smoke mean features."*

Could you discus/comment on the accuracy of AERONET retrievals, either in section 2.2 or later in the results? To what extent can we consider these the ground truth?
**R**: AERONET retrievals` accuracy is well documented (Dubovik et al., 2000a, 2000b, 2002, 2006, Sinyuk et al., 2020) and acknowledged when compared with alternative remote sensing platforms(specially satellite-based). With more than 25 years of operating a vast network of Cimel Electronique Sun–sky radiometers across the world the Aerosol Robotic Network (AERONET) has provided highly accurate, ground-truth measurements of aerosol optical optical depth and other properties (Giles et al.,2019). It has been widely used as the main reference to evaluate and validate satellites (Gupta et al, 2018) and model products (Gliß et al., 2021).

It would take a long text describing uncertainty for all AERONET inversion products, but two worth mentioning here are the uncertainties in SSA (single scattering albedo) and ASY(asymmetry parameter), two critical intensive optical properties that are retrieved by AERONET and that are related, respectively, to absorption and size of the aerosol. For AOD > 0.4 at 440 nm (or >0.2 at longer λ), SSA uncertainty ≈ ±0.03, for lower AOD, uncertainty can be ±0.05–0.07 or larger. Regarding ASY, uncertainty is about ±0.02–0.05 when AOD is high (≥0.4 at 440 nm, ≥0.2 at longer wavelengths), but can be significantly larger at low AOD.

As suggested, a brief description on the reliability and accuracy of the AERONET product is introduced in section 2.2.

Section 2.1. To me, this section felt to elaborate and sometimes irrelevant for your study. I would suggest to limit it to a short overview of what aerosol types may be expected where (and when), perhaps aided by showing (long-term averages of) MERRA column densities or something similar, without the climate information
**R**:The authors respectfully disagree with the reviewer's perspective. We believe that characterizing the study region is indeed crucial for understanding the aerosol sources and transport patterns that we aim to model. This is especially important when dealing with natural aerosols, such as dust, where emissions are primarily driven by winds.

Figure 1. Related to the previous comment, I am not sure surface elevation is the most relevant quantity to show here.
**R**: Elevation was included on the basis that it is an important aspect to understand the context of AERONET sites. For instance, coastal proximity and the depth of inland location

are important factors in discussing the nature of the air mass affecting the site, which is also influenced by its elevation. That is why we combined the site´s geographical location with its elevation to gain  a complementary perspective.

Section 2.2: Could you provide more information on the time resolution of the AERONET data, and the time periods used for the study?
R: AERONET time resolution and the periods used for this study are informed in section 2.2 (in the new version of the manuscript).

*"The interval between direct sun measurements is typically 15 minutes, but only cloud-free conditions are considered for aerosol retrievals. For sky radiance measurements up to nine times a day at the wavelengths 0.44, 0.67, 0.87, and 1.02 µm are performed(Sinyuk et al., 2020). Regarding the time period for the current study it extends from 2003 to 2023. However, due to calibration and other operational aspects, some AERONET sites present different time ranges within this period."*

Ln 213: What is the Lidar Ratio and what is it used for?
R: Lidar Ratio represents the ratio of the extinction coefficient (scattering and absorption) to the backscatter coefficient (scattering specifically at 180 degrees). It is a key parameter for identifying and categorizing different types of aerosols since light scattering depends on the size of the particle relative to the wavelength of light. Small particles, such as those from smoke plumes, usually have a high Lidar Ratio, while large particles, such as sea salt, have a small Lidar Ratio. We included a brief description of Lidar Ratio in the manuscript as follows:

*"The Lidar Ratio is the ratio of the extinction coefficient to the backscatter coefficient and is crucial for identifying different aerosol types. It reflects how light scattering varies with particle size relative to the light wavelength. Small particles, like smoke, have a high Lidar Ratio, while large particles, like sea salt, have a low Lidar Ratio."*

Ln 216: I assumed "mixed" in this context means multiple aerosols occurring in one column, irrespective of height. Does it matter for your optical properties and retrieval thereof whether at what height different aerosol species are occurring and (how) does that affect your work?
R: Mixed state in the context of aerosol particles refers to either external or internal mixing. In an external mixing state, you may have different aerosol particles, but still each particle corresponds to a specific composition or aerosol type (ex. Black-carbon, Dust, Sulphate). In an internal mixing state, a particle is a combination of different compositions or aerosol types, for instance, with a core of Black Carbon and a coated sulfate shell. This variability in mixing states introduces uncertainty into aerosol modeling. Traditional aerosol models typically describe the emission of specific aerosol types and assume a particular mixing state, most often external. They then calculate optical properties as a weighted mean of the individual aerosol components. Our approach, however, avoids any assumptions about the mixing state.

Ln 246-247: Consider introducing table 1 at the start of previous paragraph (ln. 221). Lines 221-229 feel out of place now.

**R**:Thanks, Table 1 was moved accordingly

Table 1. Am I right that you don't use aerosol optical depth? How would that be derived when your model is used? Also, would your approach also work if you only include the optical properties are used in radiative transfer calculations (SSA, ASY at different wave lengths)?
**R**: Aerosol optical depth (AOD) is an extensive variable, meaning it does not solely express the intrinsic features of aerosols but also the aerosol loading in the vertical column. Actually, the final goal of this method is to provide an aerosol optical properties model (including SSA and ASY) that will allow the models to improve radiative transfer calculations, via better simulations of aerosol optical depth, assuming the aerosol loading is adequately represented in the models. If climate models present a more accurate prescription of SSA, and ASY, they will better simulate aerosol optical depth and, consequently, better simulate radiative transfer.

Ln 296: What exactly is the Optical Model in this context?
**R**: Optical Model here refers to typical (climatological) spectral values of aerosol intrinsic optical properties used in atmospheric radiative transfer calculations. By averaging the components of each cluster obtained, we produce mean values of spectral optical properties or mean values of microphysical properties(eg. size distribution) that can be used to calculate mean spectral optical properties for the aerosol regimes. Usually, aerosol optical models are composed of spectral single-scattering albedo (SSA), asymmetry parameter(ASY), and extinction efficiency (Qext). For the sake of clarity, we replaced "optical model" with"aerosol spectral optical model". We also made a few adjustments to the sentence to improve its reading.(Ln 323 - 325, new version). We also believe that the introductory text at the beginning of this reply helps to better understand the importance and the context of optical models here discussed.

Ln 314-320: I am not sure I fully understand the equation and accompanying text, are you computing WCSS by minimising sum_1^k W(ck), or are you seeking to minimise WCSS? Also, could you explain more what you mean by Elbow?

**R**: Within-cluster sum of squares (WCSS) measures how compact or tight the clusters are. For each cluster, it calculates the distance of each point in the cluster to the cluster centroid, then squares it and sums them up. Then, the sums were added up for all clusters. If clusters are tight and well-separated, WCSS will be small, because points are close to their centroids. If clusters are loose or overlapping, WCSS will be large. The Elbow method is based on the computation of WCSS for different numbers of clusters. As the number of clusters increases, WCSS always decreases (more clusters = tighter groups). At some point, the decrease becomes less significant. The plot of WCSS against the number of clusters will display a feature similar to an "elbow". The number of clusters where the WCSS decrease becomes less significant is the one to be selected.

We try to improve the explanation of WCSS and the Elbow Method in the text (Ln 337 – 341, new version)

Ln 251-253: what do you mean with class imbalance, and what issues regarding atmospheric measurements are you addressing exactly?

R: Class imbalance refers to the unequal representation of aerosol regimes in the dataset, where some clusters (such as C2, representing only 3.48% of observations) occur much less frequently than others (such as C1, representing 37.88%). This imbalance is typical in atmospheric measurements, where extreme but radiatively important aerosol events like intense smoke episodes are rare compared to more common background conditions. Despite their low frequency, accurate prediction of minority classes like C2 is crucial for radiative transfer calculations due to their high absorption characteristics and significant impact on aerosol direct radiative forcing.

Ln 394: What is cluster stability?

R: Stability is also used as a criterion to define the number of clusters in data analysis. It helps to evaluate whether the number of identified clusters is meaningful or merely an artifact of randomness or noise. The variation in the number of clusters $k$ in k-means is used to help identify the value that yields the most stable clustering, or the value above which there is a strong decrease in stability. High stability suggests that the clusters represent genuine patterns in the data, rather than just random fluctuations. We used the stability criteria to corroborate the number of clusters identified by the Elbow Method as the ideal choice.

Part of the explanations described here were included in the text to clarify the importance of cluster stability. (Ln: 419 – 422, new version)

Ln 438: you mention marine particle scenarios here in relation to Clusters C2-C4, but marine aerosols are barely mentioned later on. Are they not important enough, or present in all regimes in similar quantities?

R: Although present in the regional atmosphere, especially in coastal areas, marine aerosols by themself do not generate high aerosol loading scenarios, and therefore high radiative forcing over the study region, as seen in dust, smoke, and urban-pollution scenarios. As the AERONET sky inversions are restricted for AOD at 440 nm higher than 0.4, it is expected that none of the clusters represent a pure or a dominant marine aerosol regime. That's why marine aerosols are not exhaustively discussed. However, as part of the background particles across the Iberian Peninsula, marine aerosol particles undoubtedly influence the clusters identified to some degree, particularly those dominant in coastal regions.

Ln 3.1: What are the mean aerosol column densities from MERRA-2 for each regime?

R: Unless we misunderstood the referee`s request, plotting average aerosol column density for each aerosol species(6) and for each regime (5) will result in a significant number of maps. This would require a proper discussion, which would make the manuscript even longer. Also, generating these maps for the annual mean would not capture the regime dynamic we seek to represent, then we would have to do it monthly(12), which again would translate into an even higher number of plots. Moreover, we are not convinced that the aerosol column density for each aerosol species would add to the discussion, as it alone cannot be translated into radiative forcing. But we evaluate to include monthly climatology of Aerosol Optical Depth(critical to radiative forcing calculations) over the Iberian Peninsula (Figure below) in the supplements to provide a perspective on the spatial and monthly

dynamic of aerosol loading in the region. We believe that this also addresses the referee`s comment regarding Figure 1. In the text where we discussed

[Figure]

**Figure R1:** Monthly mean Aerosol Optical Depth at 550 nm over Iberian Peninsula and surroundings for the period from 2003 to 2023.

Figure 4: based on what data are these size distribution computed?
**R**: The size distributions mentioned are the average of the instantaneous size distributions retrieved by AERONET from each identified cluster.  As shown in Figure 3, which includes ASY, SSA, and other properties, each cluster is composed of many instantaneous retrievals. Figure 4 displays the average size distributions for each cluster.

Ln 492-493, table 3: Where are the contest of this table discussed? Line 500 perhaps, I would refer to table 3 on that specific line.
**R**:Table 3 is introduced at line 453 (former version) and line 485 (new version), and is mentioned again at line 492 (former version) and line 524 (new version). To reinforce the contextualization of table 3, as suggested. We also referred to it at line 500 (former version) and 532 (new version).

A general comment on figures, I find the large variety in font sizes (for example, large bold text in Figure 6, small font size in Figures 11 and 12 in combination with the low dpi of some figures is distracting from their main message.
**R**: We managed to adjust and improve the font and resolution of the figures as suggested.

Ln 501: This hypothesis was already stated on line 470.

**R**: Thanks for pointing that out. To avoid repetition, as suggested, we omitted the sentence related to the comment.

Ln 508-509: "however, … , but…" consider removing either however or but?
**R**:"However" term was removed and the text adjusted. (Ln 540 new version).

Ln 535: do you have any metrics showing whether the random forests overfitting on its training data?
**R**:Thanks for this comment. Now we included metrics that allow us to evaluate overfitting. We generated classification reports for the training data to be compared with the previous report generated for the test data presented in Table 4. We created a new Table 4 that will replace the former one. These results indicate that the model generalizes well, without significant overfitting. Even for Cluster 2, which has a small number of occurrences, the model was able to maintain high precision and score. The general accuracy did not drop critically for the test data (0.70) when compared with the train dataset (0.88), another indicator of the model's ability to generalize.

The manuscript text has been adjusted to integrate the metrics and new results displayed here. Table 4 also was adjusted.

**Table 4**. Performance metrics values of the trained model applied to the test and the train(within parenthesis) dataset to predict aerosol optical regimes based on aerosol-type column mass density.

| Clusters | Precision | Recall | F1-Score | Support (N) |
|---|---|---|---|---|
| 0 | 0.62(0.89) | 0.62(0.78) | 0.62(0.83) | 394(848) |
| 1 | 0.68(0.86) | 0.70(0.94) | 0.69(0.90) | 452(1140) |
| 2 | 0.62(0.92) | 0.60(0.76) | 0.61(.83) | 62(111) |
| 3 | 0.76(0.93) | 0.73(0.94) | 0.74(0.94) | 251(439) |
| 4 | 0.68(0.91) | 0.69(0.93) | 0.69(0.92) | 185(397) |

Line 563: Could you elaborate on the "extra training", would you need extra variables, longer time series?
**R**: Sure, longer time series are an important aspect in the context of extra training, particularly when examining the C2 cluster, which has a lower occurrence compared with the other clusters. However, that is not the only aspect; training with additional predictors could also be beneficial. For example, brown carbon, an important aerosol component, is not yet available in the current MERRA-2 aerosol reanalysis products. This further discussion was included in the manuscript (Ln 595-598, new version).

Ln 573: Accuracy is not included in Table 4.
R: Well observed, thanks. Accuracy is referred to in the text and not displayed in Table 4. We adjusted the text to consider this observation. (Ln 607, new version)

Ln 580: Looking at Table 4, does C2 not have the lowest precision?
R: The reviewer is correct, thanks for pointing that out. The discussion was adjusted to describe and contextualize C2 precision properly. (Ln 614-620 new version)

Ln 590: Based on Figure 10, is organic carbon really one of the primary factors? It's relative important of organic carbon is very similar to sea salt, SO4, and SO2.
R: We rephrased the sentence to contextualize the role of organic carbon better. Although second in terms of importance, the organic carbon relative importance value is closer to sea salt and sulphate aerosol types when compared to dust. Dust occurrence, indeed, is by far the dominant predictor of the cluster. The sentence was adjusted to express this description. (Ln: 627 – 633, new version)

Ln 623/698: Could you discuss these AOD thresholds in more detail? Is your model only trained for cases with a higher AOD? What percentage of time is this threshold actually reached? I suppose that even for relatively low aerosol optical depths, you would still want optical properties to be well-represented.

R: This AOD threshold requirement is a widely recognized limitation of AERONET retrieval of aerosol properties, particularly regarding  absorption (specifically SSA). The AERONET algorithm can only retrieve SSA when the AOD at 440 nm is higher than 0.4. Consequently, all studies based on the AERONET database to prescribe aerosol optical models must adhere to this constraint, i.e., optical models are built based on AOD conditions higher than 0.4. The frequency with which this threshold is met varies from region to region and site. In the Iberian Peninsula, the threshold was achieved only between 3% and 5 %. It is important to recognize that the accuracy of an optical model is influenced by aerosol loading. For conditions of low AOD, the error in radiative transfer calculation due to  inaccuracies in optical models prescription is minimal. However, the significance of these errors increases as AOD rises. Ideally, optical properties should be accurately represented regardless of aerosol loading, but current climate models often struggle to predict their properties accurately. Therefore, enhancing the prescription of optical models for polluted scenarios is already a notable improvement.

Ln 703: what do you exactly mean with "randomly simulated following a Gaussian…"? Please elaborate.
R: We used the central value of each optical regime cluster (the mean) with a typical spread (the standard deviation), both calculated and presented in Table 3, to randomly generate normal distributions for SSA and AE.

Ln 705: "In addition … size behaviors", does this refer to the Angstrom exponent (Figure 14)?
R: Yes, we adjusted the sentence to make this clearer.

 Ln 714: "why does MERRA underestimate SSA in C3 and C4 despite the coarser particles?

**R**: There are several hypotheses for the underestimation of SSA (absorption overestimation) in MERRA-2. This issue was also highlighted in a previous study by Bakatsoula et al. (2023). One factor influencing the absorption in MERRA-2 is the lack of brown carbon in its aerosol type classification scheme. Additionally, MERRA-2 tends to underestimate SSA for other aerosol types, particularly for black carbon, as noted by Bakatsoula et al. (2023). The same study also pointed out that MERRA-2 shows excessive absorption for dust aerosol types. Therefore, the underestimation of SSA in MERRA-2 can be related to several factors. It is important to mention that despite its widespread use by the community, including this study, which indicates its importance and relevance for atmospheric studies, MERRA-2 aerosol products have inherent uncertainties that prevent them from being regarded as the absolute truth, especially regarding aerosol dynamic studies.

Ln 727:" What exactly are the expected and observed values referring to?
**R**:We are comparing the SSA and AE distribution as predicted by our model with MERRA-2. We adjusted the text to clarify as follows:

Ln 723-727: The model preserves the distribution of optical properties of less frequent aerosol regimes while capturing MERRA-2 features without the need for explicit class imbalance treatment, with C2's highly absorbing and dominant fine mode conditions reflected in both SSA and AE predictions, with the distributions of values across clusters showing coherence with MERRA-2 values.

Figure 12: I am not sure if a diverging colormap is the most appropriate choice here.
**R**:Thanks, we tried an alternative colormap in order to better depict the SSA variability.

Figures 13-15: Do the dashed lines show the means? If so, please state in the caption.
**R**:Yes, thanks, stated in the caption.

Ln 784-786: could you elaborate on the generalizability of your approach and how your approach would benefit atmospheric models, e.g. climate simulations? Would you need to many random forests, each of a small part of the world, and use those to infer spectral optical properties given column amounts of the various aerosol species?
**R**: As described, we present an alternative approach for atmospheric aerosol/climate modeling communities to prescribe aerosol spectral optical properties(optical models) in climate models. With a decision criteria based on the predicted combination of mass loading of aerosol species(Dust, Sea salt, Black carbon etc.) in each grid point column, our prescription strategy search to produce a more realistic spatial and temporal dynamic in intensive optical properties in climate models required to calculate Aerosol Optical Depth(AOD), the primary variable to the simulation of aerosol Direct Radiative Effect. This is an important advance compared to the conventional prescription of optical models based on static areas of influence and on seasonality of aerosol sources emission (ex. Hoelzemann et al., 2009, Rosario et al., 2013, Chen et al., 2024). This approach can also contribute to understanding the results of climate models that, instead of prescribing an observed-based

intensive optical model, try to explicitly predict optical properties based on assumptions such as the state of mixture assumption (externally and internally).

Regarding the application of random forest: once implemented in the climate model, the model trained using random forest to prescribe the optical models will require to be fed with the combination of aerosol types column mass loading predicted by the climate model. Once the decision of the likely cluster is made, the corresponding spectral optical model will be taken from a look-up table and applied in radiative transfer calculations. The approach can rely either on look-up tables of global or regional optical models. Therefore, the model training to prescribe optical models can be done on a global or regional basis.

Ln 785: "aerosol radiative forcing" would it be possible to also illustrate the possible benefit your approach in terms of aerosol radiative effects? For example, using clear sky radiative transfer computations using either your aerosol optical properties or those from MERRA-2 aerosol optical properties.

R: We appreciate the reviewer's suggestion; however, the inclusion of aerosol radiative forcing analysis and its intercomparison between our calculations and MERRA-2 would be a new work in itself, as it would require a comprehensive analysis and deeper discussion. In this paper, we focus on describing the developed approach and on evaluating its ability to produce optical properties prescription coherent with MERRA-2 predicted optical properties. With our approach, we not only achieved this but also identified differences between our and MERRA-2 optical properties prescriptions that are consistent with previous studies that evaluated MERRA-2 against AERONET (Bakatsoula et al., 2023).

**Minor comments & technical corrections**

Ln 16. "simulation aerosol" -> "simulation of aerosol"
R: Thanks, replaced

Ln 20. "a observational-based" -> "an observation-based"
R: Thanks, corrected

Ln 37. "Aerosol particles' importance", maybe rephrase to "The importance of aerosols"
R: Thanks, rephrased

Ln 38. What do you mean with "direct players"
R: The sentence (Ln 38-40) was adjusted to better express the meaning of direct players.

Ln 40. Could you be more specific with this sentence, it is not very clear to me
R: We believe that with the adjustment made in the previous sentence and a minor adjustment (inclusion of the term "participation") the sentence is now clearer. (Ln 39-42)

Ln 60. "due to treatment of aerosol mixing state" can you formulate more clearly?

**R**: Aerosol particles can be found in the atmosphere externally mixed (each particle represents a specific chemical composition) and/or internally mixed (one particle is a combination of particles with different compositions). While all aerosols scatter light, only a few species are significant absorbers of incoming solar radiation. These absorbing aerosol species include Black-Carbon(BC), dust, and absorbing organic aerosol (called brown carbon (BrC)). Condensable gas-phase species create a secondary aerosol coating of predominantly scattering material that either fully or partially coats these absorbing aerosols, this is usually how the internal mixing state is produced. In the case of BC, secondary and primary coatings can enhance absorption of light. Climate models struggle to simulate properly an aerosol mixing state, several just assume external mixing, which is not often the case. While some try to simulate somehow the internal mixing. Brown et al. (2021) point out that idealized internal mixing assumptions used in climate models tend to overestimate BC absorption enhancement when compared to observations. This is just one of the aspects related

We reformulated the sentence(Ln 70 new version):

Ln 70. "microphysical" missing word?

**R**: Yes, thanks. We added properties, so now it reads "*microphysical properties*" (Ln 82 new version)

Ln 102-103. "aerosol scenarios variability", should "variability" be omitted here?

**R**: Yes, it sounds better, thanks!. Variability term omitted. (Ln 115 new version)

Ln 241. When or where?

**R**: We omitted "where". (Ln 258 new version)

Ln 336-337:   "data fitted to a training process,…" could you rephrase more clearly

 Ln 339-341: "mass density, trying to" This sentence does not flow well.

**R**: Considering the reviewer comments on both lines references, we rewrote the entire sentence, from Ln 336 to 341(old version), as follows:

*"With this, we built a time series of collocated MERRA-2 aerosol types of column mass density with the developed clusters occurrences over each AERONET site, which was used in a training process aiming to predict the suitable cluster given a specific combination of aerosol types of column mass density predicted."*

We believe that the sentence is clearer. (Ln: 358-362 new version)

Ln 367: "takes .. account" is redundant.

**R**:Thanks, "into account" was omitted (ln 338 new version)

Figure 3 misses (a) (b) (c) (d) labels
R: Thanks, labels included.

Figure 9: The colorbar misses a label
**R**: Thanks, colorbar label included

Ln 582: What do you mean with "cost"?
**R**: We understand that replace cost would provide better understanding, so the sentence was adjusted to

*"mislabeling this aerosol regime would translate in higher radiative error"*

Ln 635: What do you mean with "corridor"?
**R**: Corridor meant the path (or way). We replace "corridor" with "path" so it can be clearer.
(Ln: 652 new version).

Ln 658: "Regimes regarding" -> "regimes such as"?
**R**:Adjusted as suggested (Ln: 675)

Ln 682: "550 nm field" -> "550 nm"
**R**: "field" omitted. (Ln: 699 new version)

Ln 723: What do you mean with "physical distribution characteristics"?
**R**:We changed the words to clarify as follows:
Ln 723-727: The model preserves the distribution of optical properties of less frequent aerosol regimes while capturing MERRA-2 features without the need for explicit class imbalance treatment, with C2's highly absorbing and dominant fine mode conditions reflected in both SSA and AE predictions, with the distributions of values across clusters showing coherence with MERRA-2 values.

**References**

Chen, S.; Dai, C.; Liu, N.; Lian, W.; Zhang, Y.; Wu, F.; Zhang, C.; Cui, S.; Wei, H. A Regional Aerosol Model for the Oceanic Area around Eastern China Based on Aerosol Robotic Network (AERONET). *Remote Sens.* **2024**, *16*, 1106. https://doi.org/10.3390/rs16061106

Gliß, J., Mortier, A., Schulz, M., Andrews, E., Balkanski, Y., Bauer, S. E., Benedictow, A. M. K., Bian, H., Checa-Garcia, R., Chin, M., Ginoux, P., Griesfeller, J. J., Heckel, A., Kipling, Z., Kirkevåg, A., Kokkola, H., Laj, P., Le Sager, P., Lund, M. T., Lund Myhre, C., Matsui, H.,

Myhre, G., Neubauer, D., van Noije, T., North, P., Olivié, D. J. L., Rémy, S., Sogacheva, L., Takemura, T., Tsigaridis, K., and Tsyro, S. G.: AeroCom phase III multi-model evaluation of the aerosol life cycle and optical properties using ground- and space-based remote sensing as well as surface in situ observations, Atmos. Chem. Phys., 21, 87–128, https://doi.org/10.5194/acp-21-87-2021, 2021.

Gupta, P., Remer, L. A., Levy, R. C., and Mattoo, S.: Validation of MODIS 3 km land aerosol optical depth from NASA's EOS Terra and Aqua missions, Atmos. Meas. Tech., 11, 3145–3159, https://doi.org/10.5194/amt-11-3145-2018, 2018.

Collow, A. B., Colarco, P. R., da Silva, A. M., Buchard, V., Bian, H., Chin, M., Das, S., Govindaraju, R., Kim, D., and Aquila, V.: Benchmarking GOCART-2G in the Goddard Earth Observing System (GEOS), Geosci. Model Dev., 17, 1443–1468, https://doi.org/10.5194/gmd-17-1443-2024, 2024.

Dubovik, O. and King, M.: A flexible inversion algorithm for retrieval of aerosol optical properties from Sun and sky radiance measurements, J. Geophys. Res., 105, 20763–20696, 2000.

Dubovik, O., Smirnov, A., Holben, B. N., King, M. D., Kaufman, Y. J., Eck, T. F., and Slutsker, I.: Accuracy assessment of aerosol optical properties retrieved from Aerosol Robotic Network (AERONET) Sun and sky radiance measurements, J. Geophys. Res., 105, 9791–9806, 2000.

Dubovik, O., Holben, B. N., Eck, T. F., Smirnov, A., Kaufman, Y. J., King, M. D., Tanre, D., and Slutsker, I.: Variability of absorption and optical properties of key aerosol types observed worldwide locations, J. Atmos. Sci., 59, 590–608, 2002.

Dubovik, O., Sinyuk, A., Lapyonok, T., Holben, B. N., Mishchenko, M., Yang, P., Eck, T. F., Volten, H., Munoz, O., Veihelmann, B., van der Zande, W. J., Leon, J. F., Sorokin, M., and Slutsker, I.: Application of spheroid models to account for aerosol particle nonsphericity in remote sensing of desert dust, J. Geophys. Res., 111, D11208, https://doi.org/10.1029/2005JD006619, 2006.

Sinyuk, A., Holben, B. N., Eck, T. F., Giles, D. M., Slutsker, I., Korkin, S., Schafer, J. S., Smirnov, A., Sorokin, M., and Lyapustin, A.: The AERONET Version 3 aerosol retrieval algorithm, associated uncertainties and comparisons to Version 2, Atmos. Meas. Tech., 13, 3375–3411, https://doi.org/10.5194/amt-13-3375-2020, 2020.

Giles, D. M., Sinyuk, A., Sorokin, M. G., Schafer, J. S., Smirnov, A., Slutsker, I., Eck, T. F., Holben, B. N., Lewis, J. R., Campbell, J. R., Welton, E. J., Korkin, S. V., and Lyapustin, A. I.: Advancements in the Aerosol Robotic Network (AERONET) Version 3 database – automated near-real-time quality control algorithm with improved cloud screening for Sun photometer aerosol optical depth (AOD) measurements, Atmos. Meas. Tech., 12, 169–209, https://doi.org/10.5194/amt-12-169-2019, 2019.

Vasiliki D. Bakatsoula, Marios-Bruno Korras-Carraca, Nikolaos Hatzianastassiou, Christos Matsoukas, A comparison of atmospheric aerosol absorption properties from the MERRA-2 reanalysis with AERONET, Atmospheric Environment, Volume 311, 2023, 119997, ISSN 1352-2310, https://doi.org/10.1016/j.atmosenv.2023.119997.

Rosário, N. E., Longo, K. M., Freitas, S. R., Yamasoe, M. A., and Fonseca, R. M.: Modeling the South American regional smoke plume: aerosol optical depth variability and surface shortwave flux perturbation, Atmos. Chem. Phys., 13, 2923–2938, https://doi.org/10.5194/acp-13-2923-2013, 2013.

Hoelzemann, J. J., Longo, K. M., Fonseca, R. M., do Ros´ario, N. M. E., Elbern, H., Freitas, S. R., and Pires, C.: Regional representativity of AERONET observation sites during the biomass burning season in South America determined by correlation studies with MODIS Aerosol Optical Depth, J. Geophys. Res., 114, D13301, doi:10.1029/2008jd010369, 2009

---

## Author Comment (AC2)

**RC2: 'Comment on egusphere-2025-454', Anonymous Referee #2, 15 Jul 2025 reply**

The study characterizes the typical aerosol intensive optical properties affecting the Iberian Peninsula (IP), comprising Spain and Portugal, using the atmospheric column inversion products from the AERONET sites. The authors employed K-means clustering to analyze historical aerosol intensive properties across all AERONET that operated for at least 2 years and has the highest quality dataset level 2.0 available. Five distinct aerosol optical regimes affecting the IP were identified based on the clustering technique, followed by the utilization of aerosol-type columnar mass density data (dust, organic carbon, black carbon, sea-salt, and sulphates) from MERRA-2 reanalysis to predict the aerosol optical regime using the Random Forest supervised learning methodology.

The performance of the trained model was tested under various aerosol scenarios, and the predictions ranged from 60% to 75% with accuracy exceeding 90% when predicting solely dust or non-dust optical regimes. Overall, the study is very interesting and its to the journal scope. The manuscript is well-written but requires some improvement in clarity on certain aspects before re-consideration. Recent literature needs to be cited.

We truly appreciate the time and effort Anonymous Referee #2 put into reviewing our manuscript. Their thoughtful questions and suggestions helped clarify our work and offered meaningful ideas for how we might build on it in the future. We are grateful for the constructive feedback, which will undoubtedly be very helpful in shaping the next steps of our research.

**Comments:**
**Line 37:** Statement starting with 'Via'?

**Reply:** The text was adjusted to attend reviewer 01 requirement and also to avoid to start "Via" (Ln 39, new version)

**Line 70:** compositions -> composition,

**Reply:** replaced

**Line 70:** It should be 'microphysical properties'
**Reply:** corrected

**Line 70:** computations -> computation
**Reply:** corrected

**Line 76:** What parameters are being referred to in 'aerosol simulation'?
**Reply:** The text was adjusted to better described Zhong et al. (2002) results. (Ln 89, new version)

Former text was replaced for new one:

**Former text:** *"Zhong et al. (2022) used relationships from an ensemble of aerosol models and satellite observations to identify the primary source of uncertainty in aerosol modelling results. Their study pointed out the incorrect lifetimes and the underestimation of mass extinction coefficients as the most critical drivers of bias in aerosol simulations."*

**New text:** *"Zhong et al. (2022) used relationships from an ensemble of aerosol models and satellite observations to identify the primary source of uncertainty in aerosol modelling results in biomass burning regions. Their study pointed out the incorrect simulations of lifetimes and the underestimation of mass extinction coefficients as the main reasons for their difficulty in matching observed aerosol optical depth (AOD).*

**Line 197:** What do you mean by observation-constrained approaches? Are you referring to the threshold based aerosol type classification methods? Please clarify.
**Reply:** The term "observation-based approaches" better describes what we meant, so we adjusted the text accordingly.
Traditionally, climate models assume, constrained by computational capacity, a limited set of aerosol species, including organic carbon, black carbon, dust, sulfate, and marine aerosols. These models explicitly resolve the transport and removal processes of these species, treating them as an external mixture. Therefore, the optical properties involved in radiative interactions are calculated as a mass-weighted average of individual species at each grid point. The assumption of external mixing may not always be accurate, leading to significant uncertainties, such as excessive absorption by smoke aerosols and inaccuracies in dust size fractions.
Observation-based approaches, such as those provided by AERONET retrieval climatology, attribute intensive optical properties to an effective aerosol based on actual observations. This method aims to reduce the uncertainties arising from the explicit simulation of these properties in climate models. The Iberian Peninsula, influenced by a variety of aerosol types—including dust, smoke, urban-industrial emissions, and marine aerosols—presents an interesting region to test this hypothesis.

We've adjusted the text to make the idea of "restricted observational approaches" clearer (Ln 216-218).

**Lines 211-215**: What is the rationale for choosing these aerosol intensive properties? How is Lidar Ratio (LR) and Linear Depolarization Ratio (LDR) derived with AERONET sky radiance measurements? How reliable are the LR and LDR derived from AERONET?

**Reply:** This set of aerosol intensive properties is expected to capture most of the important aspects that differentiate the distinct aerosols optical regime that affect the study region. For instance, the imaginary part of the  complex refractive index and single scattering albedo (SSA) are properties indicated to separate highly absorbing aerosol regimes from moderate and low absorbing regimes. Angstrom Exponent (AE) and Asymmetry Parameter (ASY) are properties that help separate aerosol regimes characterized by distinct size distribution. LR is highly sensitive to size and composition-related information, for instance,  the real part of the complex refractive index. Meanwhile, LDR has high sensitivity to particle morphology, and it is widely used to separate dust particles from other aerosol types. The combination of LR x LDR has proved to be very helpful to categorize aerosol regimes(Groß et al., 2012*).

AERONET retrievals of LR and LDR are based on Dubovik et al (2006). Below is presented a summary based on Shin et al. (2018), which evaluated AERONET version 3 products evaluation  focusing on LR and LDR.

For each AERONET observation, the elements F11(r,n) and F22(r,n) of the Müller scattering matrix (Bohren and Huffman, 1983) are computed from the particle size distribution and the refractive index that have been inferred from the AERONET inversion product. The element F11(r,n) is proportional to the flux of scattered light in the case of unpolarised incident light, while F22(r,n) strongly depends on the angular and spectral distribution of the radiative intensity (Bohren and Huffman, 1983) as measured with AERONET's instruments (Dubovik et al., 2006). From the element F11(r,n) at the scattering angle of 180∘ and the concurrently inferred single-scattering albedo (SSA), the lidar ratio can be computed.

$$S^p_\lambda = \frac{4\pi}{\omega_\lambda F_{11,\lambda}(r,n,180°)}.$$

The calculation of the particle linear depolarisation ratio requires knowledge of the elements F11(r,n) and F22(r,n) at a scattering angle of 180°:

$$\delta^p_\lambda = \frac{1 - F_{22,\lambda}(r,n,180°)/F_{11,\lambda}(r,n,180°)}{1 + F_{22,\lambda}(r,n,180°)/F_{11,\lambda}(r,n,180°)}.$$

Several other assumptions affect AERONET quality, for instance, external mixture. However, according to Shin et al. (2018) study, the quality of the retrievals of LR and LDR from AERONET have increased for the network products version 3(used in this study), LR and LDR  show an overall improvement by moving to more realistic values when compared with previous evaluation(Müller et al., 2012).

**Line 235**: Which climate models are being referred here?
**Reply:** Climate models in general, but more specifically the CMIP6 models (Zhao et al., 2022). This was included in the text in order to be more specific.(Ln: 258, new version)

**Table 1:** Are these VMR-F, VMR-C, STD-F, STD-C, Reff-F, Reff-C provided by the AERONET inversion products or these are derived by the authors? Please clarify. Since these intensive properties are inversion products of AERONET, how did you account for their uncertainty impacting the aerosol optical regimes identified through K-means clustering (Section 2.4)? There is no much discussion on the influence of the observational/inversion uncertainty of aerosol intensive properties on the identified clusters and interpretation of your results.

**Reply:** The mentioned variables are the AERONET inversion products. We adjusted the title of Table 1 to make it more straightforward.

The reviewer is correct; all AERONET inversion products are subject to uncertainty. There are the traditional variables that have been exhaustively applied and evaluated in the literature, like SSA, and others included in recent years, like LDR, that are still subject to further evaluation to quantify their uncertainty and reliability better. Nevertheless, AERONET data undoubtedly has the best spatial and temporal coverage worldwide. Therefore, balancing data availability and uncertainties, we chose to proceed with our work.

In this study, we did not consider how the different degree of uncertainty that characterizes each inversion product influences the clustering performance. It would be indeed an interesting exercise for the next phase of our work. We need to consider how to approach this task, given the diversity of variables used, and it requires a much deeper discussion. Nevertheless, the clustering process results consistently capture in aerosol regimes that are captured by previous works in the study region (Cachorro et al., 2016; Gómez-Amo et al., 2017). So, we believe that our results are solid for this initial exploration of the method. We will include a discussion in the text addressing the impact of AERONET inversion products uncertainty on clustering methods performance, identifying this as a possible limitation that warrants further evaluation in future research.

**Line 286**: Use Sulphate or sulfate consistently throughout the manuscript.
**Reply:** Thanks, sulfate was adopted throughout the manuscript.

**Lines 285-290**: It was mentioned that the MERRA-2 Aerosol Diagnostic Product (ADP) for aerosol types is considered in this study. Dust, Black Carbon, Organic Carbon, Sea-Salt and Sulphate aerosol mass concentration at specific levels are integrated in the entire atmospheric column to obtain columnar aerosol optical properties such extinction, scattering and absorption optical depth. It is not clear on how the mass concentrations of individual species are converted to optical depths. At least proper citation of references to the method adopted might have been included. At which wavelength these are obtained? Did you validate extinction optical depth derived from MERRA-2 with the aerosol optical depth from AERONET? Similarly, how does the SSA from MERRA-2 validate with the corresponding SSA from AERONET?

**Reply:** In this part of the manuscript, we aimed to describe the aerosol variables available in Merra-2 Aerosol Diagnostic Product and provide an example of their potential to derive additional information about aerosol. We did not perform the calculations inquired by the reviewer (extinction, scattering and absorption optical depth); these are by-products of running the MERRA-2 reanalysis system and made available to the community via MERRA-2 ADP. We agree that the inclusion of the citation reference for MERRA-2 ADP (Buchard et al., 2017*) is in order, so we added that in the revised manuscript.

\* Buchard, V., and Coauthors, 2017: The MERRA-2 Aerosol Reanalysis, 1980 Onward. Part II: Evaluation and Case Studies. J. Climate, 30, 6851–6872, https://doi.org/10.1175/JCLI-D-16-0613.1.

The Merra-2 system provides optical properties at multiple wavelengths, but 550 nm is used as a reference to provide aerosol optical properties information in Merra-2 ADP.

In this study, we did not aim to validate Merra-2 AOD as such analysis falls outside the primary goal of this paper and would require a more comprehensive approach. Moreover, several studies have already compared between Merra-2 data and AERONET retrievals worldwide, highlighting both the strengths and challenges aspects. In this study, we primarily focus on what we refer to as intensive aerosol optical properties (ex. SSA). These properties are independent of the aerosol loading and depend solely on the characteristics of the aerosol itself. Both extinction and scattering optical depth depend on the amount of aerosol.
Regarding SSA, we did compare and discuss our results with Merra-2 (Topic 3., Figure 13, 14, 15, 3)

**Line 309**: There exist several methods and indices to decide on the appropriate number of clusters such as Elbow, Silhouette, Davies Bouldin, and Calinski-Harabasz indices. I have noticed that in the following study: https://doi.org/10.1016/j.atmosres.2022.106518, the authors have stated that the correct number of clusters derived from different approaches may not lead to a single solution. What is the rationale for adopting the Elbow method, except the fact that it is a widely used method for determining the optimal number of clusters?
**Reply:** The referee is correct; there are several methods to decide the number of clusters, and we can also add the Silhouette Method to the list. Depending on the method selected, one can end up with a different number of clusters. Indeed, among the cited methods, there are those more rigorous than the Elbow Method, which is regarded by its simplicity. To reduce subjectivity, we combined Elbow and Stability methods to evaluate the optimal cluster number that best captures the diversity of aerosol regime affecting the study region. As illustrated on Figure 2, the results from the stability and elbow methods indicated potential for defining a distinct number of clusters, though we found that the number is likely to be around five in our case. Based on our clustering results and our experience with optical properties regime studies, we believe that the decision based on the combined methods effectively identified five optical properties regimes that characterizes coherently the variability of aerosol regimes affecting the Iberian Peninsula. Nevertheless, apart from its positive aspects, we agree it is valuable to include in the manuscript a brief description of the limitations that characterize Elbow Methods(described below).

*"The Elbow method has been widely used because of its straightforward approach to estimating the most appropriate number of clusters. However, it still carries a certain degree of subjectivity, as it relies on visual interpretation. To reduce this subjectivity, we combined the Elbow and Stability methods to evaluate the optimal number of clusters that best represent the major aerosol regimes affecting the study region. Although more rigorous methods are available in the literature, defining the number of clusters remains a challenge,*

*and different approaches often lead to distinct solutions (Krishnaveni et al., 2023). Despite the Elbow Method limitations, the number of clusters identified in our study seems to provide a coherent  characterization of optical regimes  affecting the Iberian Peninsula"*

**Lines 325-328**: What do you mean by 'clusters average'?
**Reply:** After defining the number of clusters (number of optical properties regimes) and  the clustering process is performed, we end up with five (5) individual clusters.  Each one is characterized by sets of AERONET instantaneous retrievals of optical and microphysical properties that are expected to express its optical regime. Each AERONET  instantaneous retrieval is tagged with the cluster number that it belongs to. By averaging the properties of each cluster,we  aim to produce a typical optical and microphysical properties set of values that represent the mentioned aerosol optical regimes.

**Lines 334-336**: It was not mentioned anywhere how the times were synchronized between the AERONET inversion parameters and MERRA-2 data of aerosol species column mass density. Each of the AERONET inversion parameters and MERRA-2 aerosol species column mass densities might have different ranges of variability and units. How is this accounted for in the ML model while identifying the clusters? I mean to ask if the ML model does any scaling and normalization of different parameters. If not, won't the range of variability and units have any impact on the aerosol classification?

**Reply:** AERONET retrievals (each one tagged with its cluster reference number) and Merra-2 column mass density are synchronized considering their proximity in time. Merra-2 column mass density of aerosol types is available with a frequency of 1 hour, while AERONET instantaneous retrievals are provided with irregular time, due to its dependence on cloud cover and AOD criteria (AOD440 nm > 0.4). So, for each AERONET retrieval, our script searches for the MERRA-2 closest hour to synchronize the two dataset.

**Line 461**: Large radius spread for C3 ... What does this infer?
**Reply:** A larger fine-mode radius (C3 case) generally means the fine particles have grown (water uptake, aging, coagulation) or that the aerosol regime mixture includes sources that naturally produce slightly larger fine particles (e.g., smoke vs. fresh urban soot). It usually indicates more aged, more hygroscopic, or more humidified aerosol compared to freshly emitted, dry fine particles.

**Lines 505-506**: There is no mention about categorization of seasons till now. How are months categorized into seasons?
**Reply:** We chose to analyze the cluster's temporal occurrence as a function of month (Figure 7) instead of seasons (ex. June-July-Aug; December-January-February) which is why we did not categorize the seasons. Where is reading "seasons" in the sentence (Ln 5050-506) should be "months". We corrected in the revised manuscript.

**Table 3**: What does the values in brackets correspond to? Standard deviation or error? This may be mentioned in the table caption.

**Reply:** Yes, the values in the brackets correspond to standard deviation, this description was included in the table caption. Thanks.

**Line 555**: How can you say that this would not introduce a substantial error in the radiative effect calculations? In terms of what metrics radiative effect is calculated? Radiative forcing or heating rates? Better to quantify this error. I suggest you to check this study: https://doi.org/10.1016/j.jqsrt.2024.109179, and see if this might provide some insights on the errors associated with direct radiative effects.

**Reply:** The idea behind this sentence is that if the classifier's confusion is between the two dust-regime models (clusters) the induced error in radiative transfer calculations would be lower than that if the confusion was between a dust and a non-dust regime, especially like C2, which is substantially different from any of the dust regime. This conclusion seems to be reasonable, even without metrics on the radiative effect. However, the reviewer pointed out an important aspect, the need for further and conclusive quantification of the radiative effect accuracy of each cluster, along with an examination of the implications of eventual misclassification. This would require an analysis not only of the columnar influence of aerosol optical properties but also of the heterogeneity of aerosol types across atmospheric layers, as stated in the suggested article. Thank you, this feedback will be very helpful for exploring the focus of a subsequent manuscript to complement the current one.

The text in the manuscript was adjusted to clarify and contextualize the expression *"introduce a substantial error in the radiative effect calculations"*.

**Lines 573-583**: It appears that these details are repeated again. Please check and avoid repetitions.
**Reply:** We were unable to find the repetition mentioned. This is the only part of the text where we discuss the results in Table 4. Some of these figures are highlighted in the abstract and conclusion, but not repeated in other parts of the manuscript.

**Line 598**: reanalyzes -> reanalyses
**Reply:** Thanks, corrected.

**Figure 10**: Short forms (dst, oac, ssl, so4, so2, bcc) used as x-axis labels for features should be defined in the figure 10 caption. It is also not clear if this relative importance is obtained for the entire IP region or the grids consisting the AERONET sites. Can you bring out similar figures to ascertain the relative importance of aerosol intrinsic parameters (Table 1) for different clusters (or aerosol scenarios) identified in this study together with the predictor variables from MERRA-2.
**Reply**: The short forms were described in the caption of figure 10. The relative importance was obtained for the grids consisting of the AERONET sites. This is now clarified in the text. If we understood well, the referee is also asking for a similar analysis of relative importance but targeting the importance of the features presented in Table 1 in the clusters prediction. We managed to calculate the relative importance of predictors(Table 1) in cluster prediction (**Figure below**) . It is possible to observe consistency between this importance score scale and the importance score scale from MERRA-2, regarding aerosol types (Figure 10). In

Figure 10, the dust (dst) mass variability emerges as the most relevant factor for determining which cluster should be applied. In the figure below, which shows the importance of the optical parameters from Table 1 for clustering, the scores are well distributed (maximum close to 0.1). However, it became clear that higher wavelengths (near infrared 870 and 1020 nm) and optical parameters(ASYmmetry parameter and Linear Depolarization Ratio),that best differentiate dust from other aerosol types, presented the highest importances.

[Figure]

We understand that this new plot results corroborate the result presented in Figure 10 and the related discussion, which show that dust variability is central to the aerosol optical regime variability over Iberian Peninsula. However, to avoid increasing the manuscript extension since it is already long, we include the figure above in the supplement and in the manuscript we briefly refer to it and to this coherence between importance score obtained for the cluster predictors(figure above) and the relative importance of MERRA-2 aerosol types as aerosol optical regime predictor.

**Line 625**: All of a sudden MERRA-2 AOD field is taken as a reference. AERONET sites also provide the AOD and SSA values, which could have been checked during the period of various scenarios (Case#01, Case#02, Case#03, Case#04).

**Reply**: We aimed to test our prescription approach in these case studies for scenarios where AOD at 440 nm meets AERONET threshold used to perform retrievals of SSA (AOD@440 nm > 0.4). However, given that the prescription is done based on a MERRA-2 map of the combination of aerosol types column density, the only way to filter areas across IP where AOD@440 nm > 0.4 is by using the AOD field from MERRA-2. AERONET provides local AOD at specific sites. This is why we took MERRA-2 AOD as a reference, and this is the context in which the sentence at line 625 was written.
We adjusted the text to make it clearer.

**Lines 673-674**: 'lower computational cost' --> How is this quantified? Have you compared with any other methods of aerosol classification?

**Reply**: Thanks. Indeed, we should better contextualize the reference to lower computational cost here. The reduction of computation cost achieved by using prescribed aerosol optical models is well recognized and undisputed, particularly when compared to the modeling systems that simulate Mie calculations online or systems that perform 3-D simulation of optical properties based on aerosol types and their mixing state, which are computationally more expensive than columnar prescriptions.

**Figure 11** Caption: MODIS Terra?

**Reply:** Corrected to MODIS Terra and Aqua since the images are selected from overpassing of both satellites. We adjusted the caption accordingly.

**Lines 697-699**: Earlier it was mentioned AERONET AOD > 0.4 but for MERRA-2 AOD > 0.3. Why?

**Reply:** This has to do with the difference in AOD as a function of wavelength. When describing AERONET threshold to provide sky retrievals such as SSA we were referring to AOD at 440 nm (0.4), here for MERRA-2 we are talking about AOD at 550 nm (0.3), that`s why the values are different. MERRA-2 ADP provides AOD at 550 nm. The MERRA-2 AOD at 550 nm threshold(0.3) goal is to provide and compare SSA only for region where aerosol loading is above the limited adopted for AERONET to perform SSA retrieval (AOD at 440 nm > 0.4)

**References**

Groß, S., Tesche, M., Freudenthaler, V., Toledano, C., Wiegner, M., Ansmann, A., Althausen, D. and Seefeldner, M. (2011) 'Characterization of Saharan dust, marine aerosols and mixtures of biomass-burning aerosols and dust by means of multi-wavelength depolarization

and Raman lidar measurements during SAMUM 2', Tellus B: Chemical and Physical Meteorology, 63(4), p. 706-724. Available at: https://doi.org/10.1111/j.1600-0889.2011.00556.x.

Dubovik, O., Sinyuk, A., Lapyonok, T., Holben, B. N., Mishchenko, M., Yang, P., Eck, T. F., Volten, H., Muñoz, O., Veihelmann, B., van der Zande, W. J., Leon, J. F., Sorokin, M., and Slutsker, I.: Application of spheroid models to account for aerosol particle nonsphericity in remote sensing of desert dust, J. Geophys. Res.-Atmos., 111, D11208, https://doi.org/10.1029/2005JD006619, 2006.

Shin, S.-K., Tesche, M., Kim, K., Kezoudi, M., Tatarov, B., Müller, D., and Noh, Y.: On the spectral depolarisation and lidar ratio of mineral dust provided in the AERONET version 3 inversion product, Atmos. Chem. Phys., 18, 12735–12746, https://doi.org/10.5194/acp-18-12735-2018, 2018.

Müller, D., Weinzierl, B., Petzold, A., Kandler, K., Ansmann, A., Müller, T., Tesche, M., Freudenthaler, V., Esselborn, M., Heese, B., Althausen, D., Schladitz, A., Otto, S., and Knippertz, P.: Mineral dust observed with AERONET Sun photometer, Raman lidar, and in situ instruments during SAMUM 2006: Shape-independent particle properties, J. Geophys. Res.-Atmos., 115, D07202, https://doi.org/10.1029/2009JD012520, 2010.

Zhao, A., Ryder, C. L., and Wilcox, L. J.: How well do the CMIP6 models simulate dust aerosols?, Atmos. Chem. Phys., 22, 2095–2119, https://doi.org/10.5194/acp-22-2095-2022, 2022.

Gómez-Amo, J. L., Estellés, V., Marcos, C., Segura, S., Esteve, A. R., Pedrós, R., Utrillas, M. P., and Martínez-Lozano, J. A.: Impact of dust and smoke mixing on column-integrated aerosol properties from observations during a severe wildfire episode over Valencia (Spain), Science Total Environ., 599–600, 2121–2134, https://doi.org/10.1016/j.scitotenv.2017.05.041, 2017.

Cachorro, V. E., Burgos, M. A., Mateos, D., Toledano, C., Bennouna, Y., Torres, B., de Frutos, Á. M., and Herguedas, Á.: Inventory of African desert dust events in the north-central Iberian Peninsula in 2003–2014 based on sun-photometer–AERONET and particulate-mass–EMEP data, Atmos. Chem. Phys., 16, 8227–8248, https://doi.org/10.5194/acp-16-8227-2016, 2016.

Krishnaveni A. Sai, B.L. Madhavan, M. Venkat Ratnam, Aerosol classification using fuzzy clustering over a tropical rural site, Atmospheric Research, Volume 282, 2023, 106518, ISSN 0169-8095, https://doi.org/10.1016/j.atmosres.2022.106518.